Cranial anatomy of Besanosaurus leptorhynchus Dal Sasso & Pinna, 1996 (Reptilia: Ichthyosauria) from the Middle Triassic Besano Formation of Monte San Giorgio, Italy/Switzerland: taxonomic and palaeobiological implications

http://orcid.org/0000-0001-6359-3067 Bindellini Gabriele 1 gabriele.bindellini@unimi.it
http://orcid.org/0000-0002-6336-8916 Wolniewicz Andrzej S. 2
http://orcid.org/0000-0002-6888-3546 Miedema Feiko 3 4
http://orcid.org/0000-0002-6301-8983 Scheyer Torsten M. 4
http://orcid.org/0000-0001-6791-6601 Dal Sasso Cristiano 5 cristiano.dalsasso@comune.milano.it
1 Dipartimento di Scienze della Terra “Ardito Desio”, Università degli Studi di Milano , Milano , Italy
2 Institute of Paleobiology, Polish Academy of Sciences , Warsaw , Poland
3 Staatliches Museum für Naturkunde Stuttgart , Stuttgart , Germany
4 Paläontologisches Institut und Museum, Universität Zürich , Zürich , Switzerland
5 Sezione di Paleontologia dei Vertebrati, Museo di Storia Naturale di Milano , Milano , Italy
Young Mark
Electronic publication date: 2021 May 6
Publication date: 2021
Volume: 9
Electronic Location ID: e11179
Received 2020 Nov 4; Accepted 2021 Mar 8
Copyright: © 2021 Bindellini et al.
Copyright year: 2021
Copyright holder: Bindellini et al.
License: This is an open access article distributed under the terms of the Creative Commons Attribution License, which permits unrestricted use, distribution, reproduction and adaptation in any medium and for any purpose provided that it is properly attributed. For attribution, the original author(s), title, publication source (PeerJ) and either DOI or URL of the article must be cited.
License URL: https://creativecommons.org/licenses/by/4.0/

Keywords: Ichthyosauria, Shastasauridae, Middle Triassic, Besano Formation, Monte San Giorgio, Cranial anatomy, Osteology, Phylogeny, Longirostry, Marine reptiles

Funding: Natural Environment Research Council (NERC) (NE/L501530/1) Department of Earth Sciences, University of Oxford (2013–2017) Swiss National Science Foundation (31003A_179401) Comparative data collection by Andrzej S. Wolniewicz was funded by a Natural Environment Research Council (NERC) Ph. D. Studentship (cohort grant NE/L501530/1) carried out at the Department of Earth Sciences, University of Oxford (2013–2017). Torsten M. Scheyer was supproted by the Swiss National Science Foundation (grant no. 31003A_179401). The funders had no role in study design, data collection and analysis, decision to publish, or preparation of the manuscript.

==============================
Besanosaurus leptorhynchus Dal Sasso & Pinna, 1996 was described on the basis of a single fossil excavated near Besano (Italy) nearly three decades ago. Here, we re-examine its cranial osteology and assign five additional specimens to B. leptorhynchus, four of which were so far undescribed. All of the referred specimens were collected from the Middle Triassic outcrops of the Monte San Giorgio area (Italy/Switzerland) and are housed in various museum collections in Europe. The revised diagnosis of the taxon includes the following combination of cranial characters: extreme longirostry; an elongate frontal not participating in the supratemporal fenestra; a prominent ‘triangular process’ of the quadrate; a caudoventral exposure of the postorbital on the skull roof; a prominent coronoid (preglenoid) process of the surangular; tiny conical teeth with coarsely-striated crown surfaces and deeply-grooved roots; mesial maxillary teeth set in sockets; distal maxillary teeth set in a short groove. All these characters are shared with the holotype of Mikadocephalus gracilirostris Maisch & Matzke, 1997, which we consider as a junior synonym of B. leptorhynchus. An updated phylogenetic analysis, which includes revised scores for B. leptorhynchus and several other shastasaurids, recovers B. leptorhynchus as a basal merriamosaurian, but it is unclear if Shastasauridae form a clade, or represent a paraphyletic group. The inferred body length of the examined specimens ranges from 1 m to about 8 m. The extreme longirostry suggests that B. leptorhynchus primarily fed on small and elusive prey, feeding lower in the food web than an apex predator: a novel ecological specialisation never reported before the Anisian in a large diapsid. This specialization might have triggered an increase of body size and helped to maintain low competition among the diverse ichthyosaur fauna of the Besano Formation.

Introduction

Shastasaurids were important components of Triassic marine ecosystems and represented, along with Cymbospondylidae, one of the earliest groups of medium to large-bodied ichthyosaurs. In fact, members of this varied clade ranged in size from about 6 m to more than 20 m and included the whale-sized Shonisaurus sikanniensis, the largest ichthyosaur known to date (Nicholls & Manabe, 2004). As defined by Ji et al. (2016), Shastasauridae include six genera of long-bodied (presacral count > 55) forms: Shastasaurus, Besanosaurus, Guanlingsaurus, Guizhouichthyosaurus, Shonisaurus and ‘Callawayia’ wolonggangense (Camp, 1980; Dal Sasso & Pinna, 1996; Maisch & Matzke, 1997a; Li & You, 2002; Nicholls & Manabe, 2004; Maisch et al., 2006b; Pan, Jiang & Sun, 2006; Chen, Cheng & Sander, 2007; Shang & Li, 2009; Sander et al., 2011; Ji et al., 2013). The monophyly of Shastasauridae (recovered by e.g., Ji et al., 2013, 2016; Jiang et al., 2016; Motani et al., 2017; Huang et al., 2019) has often been questioned and the clade has been recovered as paraphyletic by several authors (e.g., Maisch & Matzke, 2000; Sander, 2000; Sander et al., 2011; Moon, 2017; Moon & Stubbs, 2020). The validity of some of the shastasaurid taxa has also been questioned (e.g., Guanlingsaurus: Sander et al., 2011; Ji et al., 2013; Guizhouichthyosaurus: Shang & Li, 2009; ‘Callawayia’ wolonggangense: Chen, Cheng & Sander, 2007; Ji et al., 2016). The oldest undisputed shastasaurids are known from the Anisian (Middle Triassic) and the youngest persisted to the latest Rhaetian (Late Triassic), gaining a global distribution from present-day Europe, throughout Asia, to North America (e.g., Wiman, 1910; Storrs, 1994; Dal Sasso & Pinna, 1996; Sander, 2000; Benton et al., 2013; Maxwell & Kear, 2013; Fischer et al., 2014; Lomax et al., 2018; but see Martin et al., 2015). A considerable diversity of medium- and large-bodied ichthyosaurs, morphologically similar to shastasaurids, was also reported from the Lower to Middle Triassic of Svalbard, and possibly includes some of the earliest known shastasaurids, such as Pessosaurus polaris and Pessopteryx nisseri (e.g. Wiman, 1910; McGowan & Motani, 2003; Maxwell & Kear, 2013; Engelschiøn et al., 2018). However, due to the fragmentary nature of the specimens of these taxa, their validity and taxonomic affinity remain a matter of debate. Some of the specimens previously referred to Pessopteryx nisseri are morphologically very similar to Besanosaurus leptorhynchus (McGowan & Motani, 2003), which indicates their shastasaurid affinity, but these specimens comprise postcranial material only and will therefore not be discussed here.

Ichthyosaurs are among the most abundant fossil reptiles of the UNESCO World Heritage Site of Besano–Monte San Giorgio (Lombardy, Italy; and Canton Ticino, Switzerland; Fig. 1), which is one of the richest sites for Middle Triassic marine palaeobiodiversity (e.g., Rieppel, 2019). Dal Sasso & Pinna (1996) named B. leptorhynchus and identified it as a new shastasaurid ichthyosaur on the basis of a complete specimen (BES SC 999). It was unearthed in 1993 in the Sasso Caldo quarry near Besano, from the Anisian bituminous shales of the Besano Formation. This new ichthyosaur was clearly different from either Mixosaurus cornalianus (the most common ichthyosaur of the Besano Formation outcrops) and Cymbospondylus buchseri (the only other medium to large-sized taxon known from Monte San Giorgio), and showed a close affinity with shastasaurids, despite exhibiting several morphological differences from all other shastasaurid taxa. Nevertheless, the Besanosaurus specimen of the Museo di Storia Naturale di Milano (MSNM) was not the only shastasaurid uncovered by that time from Monte San Giorgio/Besano. Two specimens, a medium-sized (PIMUZ T 4376) and a large-sized (PIMUZ T 4847) skeleton, were present in the collections of the Paläontologisches Institut und Museum der Universität Zürich (PIMUZ) since the late 1920s (PIMUZ records). Both skeletons were only briefly mentioned in 20th century literature (Kuhn-Schnyder, 1964; McGowan, 1976; Sander, 1989; Brinkmann, 1994, 1997; Cook, 1994) and the medium-sized skeleton was seemingly under study in the 1990s (Cook, 1994; Dal Sasso & Pinna, 1996; Brinkmann, 1997), but a comprehensive osteological description was never published.

Figure 1 Relevant fossil sites in the Monte San Giorgio area.

Map of the Monte San Giorgio area showing the Middle Triassic carbonate succession, the major paleontological quarries in the area (white circles), and the sites of origin of the specimens described in this paper (yellow rhombuses).

The MSNM Besanosaurus, which represents the most complete shastasaurid from the Besano Formation discovered to date, was described in part (Dal Sasso & Pinna, 1996) when it was not yet fully prepared. Subsequent lab work confirmed the remarkable preservation of the postcranial skeleton and the presence of embryonic remains (Dal Sasso, 2001, 2004). However, the interpretation of the morphology of the holotypic skull remained problematic, due to intense overlapping and diagenetic compression of its semi-disarticulated bone elements.

Right after Dal Sasso & Pinna (1996), Maisch & Matzke (1997a) described another shastasaurid specimen (GPIT 1793/1) from the Besano Formation of Monte San Giorgio, referring it to a new genus and species, Mikadocephalus gracilirostris. The authors did not cite B. leptorhynchus, possibly because they were not aware of the existence of the specimen at that time (Sander, 2000).

Later, Maisch & Matzke (2000) and Maisch (2010) maintained the distinctness of the two genera, based on examination of GPIT 1793/1 and PIMUZ T 4376, which they referred to Mikadocephalus gracilirostris, and PIMUZ T 1895, which they considered a referred specimen of B. leptorhynchus—although the holotype was not examined personally by them (Maisch & Matzke, 2000: 7). The anatomy and taxonomy of Mikadocephalus and Besanosaurus received little attention in the late 1990s–early 2000s (but see Sander, 2000: 15; McGowan & Motani, 2003: 127), and by that time the excavations of the relevant sites at Monte San Giorgio were stopped in part in Switzerland, and totally in Italian territory. Another possible shastasaurid from Monte San Giorgio was recently identified in the collections at MSNM (BES SC 1016) and prepared for this study along with PIMUZ T 1895, which helped to clarify the anatomy of the specimens studied here.

Here, we revise the cranial anatomy of the holotype of B. leptorhynchus and describe for the first time the skulls of all other shastasaurid specimens from the Besano Formation. We further compare these materials with other ichthyosaurs and revise a previously-published data matrix to perform a phylogenetic analysis elucidating the position of the revised taxon. In addition, we address aspects of ontogenetic variation and palaeoecology of Besanosaurus.

Institutional abbreviations. GPIT, Palaeontological Collection of Tübingen University, Tübingen, Germany; GNG, Guanling National Geopark, Guanling, China; IVPP, Institute of Vertebrate Paleontology and Paleoanthropology, Chinese Academy of Sciences, Beijing, China; MSNM, Museo di Storia Naturale di Milano, Milan, Italy; PIMUZ, Paläontologisches Institut und Museum der Universität Zürich, Zürich, Switzerland; ROM, Royal Ontario Museum, Toronto, Canada; SMNS, Staatliches Museum für Naturkunde Stuttgart, Stuttgart, Germany; SPCV (currently WGSC), Wuhan Centre of China Geological Survey, Wuhan, China; TMP, Royal Tyrrell Museum of Palaeontology, Drumheller, Canada; UCMP, University of California Museum of Paleontology, Berkeley, California, USA.

Anatomical abbreviations. Osteological: ang, angular; ar, articular; at, ‘anterior terrace’ of the supratemporal fenestra; atl, atlas; avsc, impression of the anterior vertical semicircular canal; ax, axis; bas, basisphenoid; bo, basioccipital; boc, basioccipital condyle; bop, basioccipital peg; bptf, basipterygoid facet; c, centrum; cl, clavicle; cop, coronoid (preglenoid) process of the surangular; d, dentary; dg, dental groove; ds, dental socket(s); dv, dividing ridge; eca, extracondylar area; en, external naris; epi, epipterygoid; exo, exoccipital; fm, foramen magnum; fmf, foramen magnum floor; fr, frontal; hsc, impression of the horizontal semicircular canal; hy, hyoid; hyf, hypoglossal foramen; icf, internal carotid foramen; icl, interclavicle; ipv, interpterygoid vacuity; j, jugal; jcr, jugal caudal ramus; jmd, jugal medial depression; jrr, jugal rostral ramus; la, lacrimal; m, maxilla; mcr, caudal ramus of the maxilla; mh, medial head of the opisthotic; mrr, rostral ramus of the maxilla; n, nasal; na, neural arch; np, notochordal pit; nuf, (remnant of a) neurovascular foramen; nvf, neurovascular foramen; opi, opisthotic; pa, parietal; pal, palatine; pas, parasphenoid; pas cup, cultriform process of the parasphenoid; pbs, parabasisphenoid; pcop, paracoronoid process; pf, parietal foramen; pm, premaxilla; pnp, postnarial process of the maxilla; po, postorbital; pop, paroccipital process; pra, prearticular; prf, prefrontal; pro, prootic; pte, pterygoid; ptf, postfrontal; pvsc, impression of the posterior vertical semicircular canal; q, quadrate; qc, quadrate condyle; qj, quadratojugal; qjce, quadratojugal covered edge; qjf, facet for the quadratojugal; qjqf, quadratojugal facet for the quadrate; sang, surangular; sbp, subnarial process of the premaxilla; sc, scleral plate; scr, sagittal crest; soc, supraoccipital; sp, splenial; spp, supranarial process of the premaxilla; sq, squamosal; st, supratemporal; sta, stapes; su, impression of the sacculus/utriculus area; tf, supratemporal fenestra; tpq, ‘triangular process’ of the quadrate; vk pas cup, ventral keel of the cultriform process; vo, vomer. “f”, behind an abbreviation, denotes an articular facet; “p”, behind an abbreviation, denotes a process. The apostrophe (‘) always indicates a left element.

Musculature: mAMEM, musculus adductor mandibulae externus medialis; mAMEP, musculus adductor mandibulae externus profundus; mAMES, musculus adductor mandibulae externus superficialis; mPSTs, musculus pseudotemporalis superficialis.

Geological setting

The Middle Triassic sedimentary succession of Monte San Giorgio consists of four different formations (Fig. 1) deposited on a carbonate platform along the western margin of the Neo-Tethys (Furrer, 1995; Röhl et al., 2001; Etter, 2002; Stockar, Baumgartner & Condon, 2012). Above the Anisian Salvatore Dolomite lies the 5- to 16-m-thick Besano Formation (Anisian-Ladinian boundary), from which the greatest part of the well-known vertebrate fauna of Monte San Giorgio has been recovered (Bürgin et al., 1989; Furrer, 2003).

The Besano Formation (also known as Grenzbitumenzone) consists of an alternation of variably laminated dolomitic banks and bituminous shales, and sparse cineritic tuffs that are dated as late Anisian–early Ladinian (Brack & Rieber, 1986, 1993; Mundil et al., 1996; Brack et al., 2005; Wotzlaw, Brack & Storck, 2017). It was deposited in a marine setting with an estimated depth of 30–130 m (Bernasconi, 1991, 1994; Bernasconi & Riva, 1993; Furrer, 1995; Röhl et al., 2001; Etter, 2002). The middle portion of the Besano Formation, which probably yielded all specimens examined herein, was deposited in an intraplatform basin (Röhl et al., 2001) and is characterized by organic-carbon rich layers, with well-preserved macro-lamination testifying very quiet hydrodynamic conditions and a lack of post-depositional bioturbation. In all studies mentioned above, it is hypothesized that such deposition took place in a basin with mostly permanent anoxic conditions at the bottom, due to restricted water circulation. The great abundance of pelagic marine vertebrates is also typical of this portion of the Besano Formation. Recent biozonation of the Sasso Caldo site (Fig. 2), based on index-fossil invertebrates (ammonoid and the bivalve Daonella), indicate that the stratigraphic section cropping out therein is fairly consistent with the most recent biozonation reported by Brack et al. (2005) and allows confident correlation with the coeval Swiss localities (Table 1) (Bindellini et al., 2019). The holotype of B. leptorhynchus (BES SC 999) was collected at the Sasso Caldo site near Besano, below the three uppermost volcanic layers and within the N. secedensis Zone (middle Besano Formation) from stratum 65 (equivalent to layer 107 of the Mirigioli Swiss section). It is of late Anisian age and therefore the taxon represents the stratigraphically oldest shastasaurid (sensu Ji et al., 2016) known to date. The skull BES SC 1016 was collected at Sasso Caldo as well, from stratum 70 (equivalent to layer 96 of the Mirigioli Swiss section). The Swiss specimens were extracted from the N. secedensis Zone of Cava Tre Fontane (PIMUZ T 1895, 4847) and Valle Stelle (PIMUZ T 4376) mines (Figs. 1 and 2; Table 1). The exact horizon of origin of the holotypic specimen of Mikadocephalus (GPIT 1793/1) is unknown.

Figure 2 Stratigraphic log of the Besano Formation.

Stratigraphic log of the Besano Formation at the Mirigioli/Punkt 902 outcrop, with the known stratigraphic positions of the specimens described. The stratigraphic position of PIMUZ T 1895 and GPIT 1793/1 is more uncertain, thus expressed by a range line. Log modified from Brack et al. (2005); dating (in millions of years) of layer 71 from Mundil et al. (1996); dating of Tc Tuffs (layers 66–68) from Wotzlaw, Brack & Storck (2017).

Table 1 Localities and horizons.

Collection/specimen		Locality	Nat.	‘stratum’ (IT)	Layer (CH)	
MSNM	BES SC 999	Sasso Caldo	I	65	[107]	
BES SC 1016	Sasso Caldo	I	70	[96]	
PIMUZ	T 4376	Valle Stelle	CH	[82]	71	
T 4847	Cava Tre Fontane	CH	[60]	116	
T 1895	Cava Tre Fontane	CH	N. secedensis Zone	
GPIT	1793/1	Unknown CH site	CH	Besano Formation	
Note:

Localities and horizons for each specimen described in this paper. “I” (Italy) and “CH” (Switzerland) indicate the nationality of the site. ‘Stratum’ or ‘Layer’ in squared brackets [*] was deduced following Bindellini et al. (2019), since two different methods have been used to number the layers during excavations.

Materials and Methods

Preservation of the studied material

All studied specimens lay in a single bedding plane and are variably compressed by diagenetic alteration (Figs. 3–9). The skeletons embedded in the most bituminous layers (BES SC 999, PIMUZ T 1895) demonstrate better preservation but also higher bone deformation due to more extreme diagenetic compression. The specimens embedded in bituminous dolomite (PIMUZ T 4847, BES SC 1016) show less detail but are less compressed. The preservation of specimens PIMUZ T 4376 and GPIT 1793/1 is intermediate, leading to a combination of good bone preservation and limited deformation.

Figure 3 The most complete skeletons of Besanosaurus leptorhynchus.

The most complete skeletons referable to Besanosaurus leptorhynchus. (A) PIMUZ T 1895; (B) BES SC 999; (C) PIMUZ T 4376 (with a Mixosaurus specimen above it); (D) PIMUZ T 4847. Scale bars represent 50 cm.

Figure 4 Skull and mandible of Besanosaurus leptorhynchus holotype.

Skull and mandible of Besanosaurus leptorhynchus holotype BES SC 999, and their interpretative drawings. Grey dashed lines and grey labels indicate elements not visible on the surface, grey areas indicate background sediment, light grey areas indicate background bone. Abbreviations: see text. Scale bar represents 10 cm.

Figure 5 Skull and mandible of PIMUZ T 4376.

Skull and mandible of PIMUZ T 4376, and their interpretative drawings. Grey dashed lines and grey labels indicate elements not visible on the surface, grey areas indicate background sediment, light grey areas indicate background bone. Abbreviations: see text. Scale bar represents 10 cm.

Figure 6 Skull and mandible of PIMUZ T 1895.

Skull and mandible of PIMUZ T 1895, and their interpretative drawings. Grey dashed lines indicate portions of missing elements preserved as counterprints, grey areas indicate background sediment, light grey areas indicate background bone. Abbreviations: see text. Scale bar represents 10 cm.

Figure 7 Skull and mandible of PIMUZ T 4847.

Skull and mandible of PIMUZ T 4847, and their interpretative drawings. Grey areas indicate background sediment. Abbreviations: see text. Scale bar represents 10 cm.

Figure 8 Skull and mandible of GPIT 1793/1.

Skull and mandible of GPIT 1793/1, and their interpretative drawings. Grey dashed lines and grey labels indicate elements not visible on the surface, grey areas indicate background sediment, light grey areas indicate background bone. Abbreviations: see text. Scale bar represents 10 cm.

Figure 9 Skull and mandible of BES SC 1016.

Skull and mandible of BES SC 1016, and their interpretative drawings. Grey areas indicate background sediment, light grey areas indicate background bone. Abbreviations: see text. Scale bar represents 10 cm.

Disarticulation is more common in the forefins than in the hindfins, and in the post-sacral axial skeleton (BES SC 999, PIMUZ T 4376, PIMUZ T 4847). One specimen contains embryonic and soft tissue remains (BES SC 999); the largest specimen (PIMUZ T 4847) contains a large nodule in the cranial half of the thoracic region, possibly related to visceral soft tissue.

BES SC 999. The holotype of B. leptorhynchus measures 5.065 m from the tip of the rostrum to the last caudal vertebra. The skeleton is virtually complete and lies in a ventrodorsal position with the paired elements symmetrically flattened along the left and right sides of the body. In the presacral region the vertebrae and the rib cage are nearly as articulated as in vivo; in the caudal region, the elements associated with the vertebral centra (neural spines and chevrons) are relatively close to their in vivo position, but slight wave action may have displaced them to some degree. The skull is detached and set at an approximately right angle in relation to the body, and mostly exposed in left lateral view. The original fossil of BES SC 999 is stored in a special climate-controlled cabinet within the MSNM collections, still divided in 25 slabs (Dal Sasso & Pinna, 1996: fig. 5); a recomposed cast of the skeleton and the surrounding matrix is on display in the MSNM paleontological hall n°5.

PIMUZ T 4376. The best-preserved specimen, it measures 2.12 m from the tip of the rostrum to the last preserved caudal vertebra (the distalmost caudals are missing). The skull and most of the presacral axial skeleton are exposed in right laterodorsal view, and are mostly articulated; the rest of the vertebrae and ribs, as well as the limb bones, are disarticulated and variably clustered on both sides of the vertebral column.

PIMUZ T 1895. Incomplete and mostly unprepared skeleton, lacking most of the tail and the limbs, except for the tailbend (preserved in a separate slab) and the very proximal elements (girdle bones). The specimen is exposed in left laterodorsal view and semi-articulated. The preserved presacral length is around 1.40 m. The associated skull is preserved on a separate slab and the tip of the rostrum was recently prepared and transferred from a minor counterslab to the main slab for this study.

PIMUZ T 4847. The largest specimen, it has a presacral length of 3.28 m but lacks most of the post-sacral skeleton and most of the limb bones. In life, this individual likely reached a length of about 8 m, being one of the biggest among Middle Triassic ichthyosaurs. It is exposed in left lateroventral view, and only the rib cage is preserved semi-articulated. All other bones are scattered along the body profile, and only the longest skull elements retain some original orientation. The inferior quality of preservation of this specimen is partially related to the more difficult preparation of the hard dolomite layers.

GPIT 1793/1. This specimen is the holotype of Mikadocephalus gracilirostris (Maisch & Matzke, 1997a). It is preserved in three slabs that can be easily reassembled in their original position. It consists of a partial skull and lower jaw, both almost entirely disarticulated, with the exception of the dorsal bones of the dermal skull (see the description below for further information). These are exposed in ventral (internal) view. The specimen has undergone some diagenetic compression but is still relatively well preserved in 3D in comparison to the rest of the material described herein.

BES SC 1016. Collected in several elongate slabs fractured perpendicular to the rostrocaudal axis of the skull and reassembled during preparation, this specimen consists of a partially disarticulated skull and lower jaw exposed in left laterodorsal view, missing the tip of the rostrum and the caudal portion of the occipital region. The bone elements are embedded in a hard layer of dolomite, show medium–low diagenetic compression, and are crossed by several lines of fracture.

The cranial material

BES SC 999. The skull of BES SC 999 (Fig. 4) is preserved mostly on slab n°1, but a few fragmentary bones are also preserved on slab n°2 (see slab numbers in Dal Sasso & Pinna, 1996: fig. 5). The entire skull is extremely compressed measuring only 15 mm in mediolateral thickness and it is fossilised in a very peculiar position: the rostrum, the left orbit, left postorbital, and the bones of the left lower jaw are well-exposed in lateral view and are still semi-articulated whereas the rest of the bones have been displaced and rotated in a puzzling way.

In detail, part of the left maxilla is covered by fragments of the left and right nasal bones, obscuring the caudal end of the external naris. Part of the nasals, the frontals, the parietals, the right prefrontal, the right postfrontal, and the supratemporals have been overturned relative to the other elements, exposing their internal surface. Due to this displacement of the skull roof, the right side elements are now adjacent to the left ones, which are exposed in lateral (external) view. The right nasal is still partially articulated with the right frontal, whereas the left nasal is no longer articulated with the rest of the skull, being caudally separated from the left frontal by the left prefrontal. The long rostral processes of the nasals are deformed, fractured, and partly hidden in between the two premaxillae.

The left jugal is rotated relative to the orbit so that its medial surface is exposed. It now occupies a position above the left orbit, so that in the previous description of B. leptorhynchus it was misinterpreted as the prefrontal-postfrontal complex (Dal Sasso & Pinna, 1996). Likely, the jugal has been displaced in this position as a result of the disarticulation and displacement of the skull roof. Subsequently, the rostral portion of the jugal has been covered and then compressed below a disarticulated fragment of the left ?nasal.

The quadrate is exposed in caudolateral view; the basioccipital, the right stapes, the putative right opisthotic, and right exoccipital are exposed in occipital (caudal) view and clustered close to one another.

Several bones are hidden under the elements exposed on the surface. These include the left stapes, the right prefrontal and postfrontal (located under the right parietal and frontal), a small part of the basioccipital, the right squamosal, still articulated with the right jugal, the right elements of the lower jaw, the right maxilla, the right premaxilla, and the palatal elements (vomers, palatines and pterygoids). A great portion of the right nasal is covered by the left sclerotic ring, the left jugal, and the left nasal, but it is exposed caudally adjacent to the right frontal.

PIMUZ T 4376. This is the best articulated and least deformed skull (Fig. 5). Almost all elements from both sides of the skull roof are visible, thanks to a dorsal rotation that likely occurred during very slow plastic deformation of a soupy sea bottom substrate. Symmetric positions with respect to the sagittal line are retained by all bones, except for the caudal-most portion of the left ?lacrimal. The right supratemporal shows the medial process still connected to the right parietal, which preserves the sagittal crest, and the lateral portion of the bone, still in articulation, delimits the caudodorsal and the caudolateral margin of the supratemporal fenestra. The occipital area is disarticulated, with the basioccipital rotated and covering the paired elements of the left side. The right orbit and the bones bordering its caudoventral margin are slightly displaced dorsoventrally; the same has happened to the right narial opening. Only the lower right jaw is exposed and well visible in lateral view, with an unusual upside-down rotation of the detached rostral tip (rotated by 180° and curiously parallel to the right premaxilla).

PIMUZ T 1895. This specimen (Fig. 6) is mainly exposed on its left side, but is in a way very similar to PIMUZ T 4376, i.e., with the skull roof well-exposed in dorsal view, the circumnarial and orbital regions slightly compressed but visible in lateral view, and the lower jaw mediolaterally compressed and visible in lateral aspect. The bones of the latter are more displaced, so that the main elements of both sides can be seen for most of their length. Just caudal to the orbit, a vertical fracture in the bituminous matrix crosses the skull, leaving a gap of missing bone pieces. In contrast to PIMUZ T 4376, here the occipital region is not fully preserved.

PIMUZ T 4847. The skull of PIMUZ T 4847 (Fig. 7) is detached from the body, exposed in ventral view, and mostly disarticulated. The lower jaw bones lay on the two sides of the skull at some distance, partly still articulated. Unfortunately, most elements of the skull roof and the occipital region have been dispersed and lost. Both jugals are exposed in medial view and the left one is close to its original position, although missing its rostral tip. From the postorbital region of the skull only the left quadrate, the right articular and the right quadratojugal are clearly identifiable. Some bones of this specimen are impossible to identify due to its poor preservation.

GPIT 1793/1. This skull was described and designated as the holotype of Mikadocephalus gracilirostris by Maisch & Matzke (1997a). The cranial elements are mostly disarticulated and scattered on the three slabs of the bituminous matrix (Fig. 8). Despite this, the dorsal-most bones of the skull roof are still in articulation (part of the nasals, frontals, prefrontals, postfrontals and parietals), and are exposed in internal view. In addition, on both sides of the skull, the lateral portions of the prefrontals and postfrontals (i.e. supraorbital arches) are broken and dislocated, so that their dorsolateral surfaces are exposed lying on top of the internal surface of the skull roof. In our paper we propose a new interpretation of this specimen (Fig. 8 and description below), which allows for the attribution of GPIT 1793/1 to B. leptorhynchus.

BES SC 1016. This specimen lacks the tip of the rostrum and some occipital elements (Fig. 9). The skull is partly disarticulated and incomplete. Some of the right dorsal elements of the skull roof (prefrontal, postfrontal, frontal and parietal) can be recognized; they delimitate a very deformed orbit and temporal fossa. The left jugal and maxilla are preserved in lateral view and are almost complete. The left premaxilla is also preserved in lateral view but it is missing the rostral tip; at its caudal end, the supranarial and subnarial processes are clear and intact, well-defining the rostral portion of the left external naris. The left dentary is exposed laterally, whereas the left surangular is exposed in medial view; the right lower jaw is visible in dorsal view. Remarkably, this specimen preserves both pterygoids which are well exposed and still articulated with each other.

Methods

X-ray computed tomography (CT) was performed on the whole skeleton of the holotype of B. leptorhynchus and on BES SC 1016 with a Siemens Somatom Definition Dual Source CT Scanner at the Radiology Department of the Fondazione IRCCS “Cà Granda” Ospedale Maggiore Policlinico di Milano. The best CT imaging was obtained with a bone algorithm on transverse (axial) slices with 140 kV voltage and 180–270 mA current and a slice thickness of 0.3 mm (Crasti, 2019). Data were exported in DICOM format using eFilm (v. 1.5.3; Merge eFilm, Toronto, ON, Canada). Analysis and post-processing were performed with RadiAnt, 3DimViewer, and Synedra View Personal. Multiplanar reconstructions (MPR) and volume rendering reconstructions (VR) allowed to inspect the bones hidden under other ones within the matrix, otherwise impossible to study without damaging the fossil (Fig. 10).

Figure 10 CT slices of BES SC 999 postnarial region.

Selected (most informative) CT slices of the postnarial region of the holotype of Besanosaurus leptorhynchus (BES SC 999), ordered by depth (top to bottom, from the deepest to the most surface level). (A) original slices; (B) slices interpretations. Abbreviations: see text. Scale bar represents 10 cm.

We used photogrammetry to visualise the cranial anatomy of specimens BES SC 999, PIMUZ T 4376, and GPIT 1793/1 (Files S1, S2 and S3). 3D models of the skulls were obtained with Meshroom by processing around 100 shots for each specimen. Photos of all studied specimens were taken with a Nikon D3500 camera.

To test the phylogenetic position of Besanosaurus leptorhynchus we used a modified matrix (see discussion) from Huang et al. (2019). This was analysed in TNT 1.5 (Goloboff, Farris & Nixon, 2008; Goloboff & Catalano, 2016), with memory set to hold 99,999 trees. The New Technology search option (a combination of Sectorial Search, Ratchet, Drift and Tree fusing, with 100 random addition sequences) was used, followed by a round of TBR branch-swapping. Bremer support values were calculated in TNT 1.5 using the built-in Bremer Support tool.

Results: revised taxonomy of besanosaurus leptorhynchus

Mikadocephalus gracilirostris as a junior synonym of Besanosaurus leptorhynchus

Maisch & Matzke (1997a) recognized the following characters as possible autapomorphies of Mikadocephalus gracilirostris: (1) exceedingly slender and gracile snout; (2) presence of a triangular medioventral process on the quadrate; (3) a very large and well-developed coronoid process on the surangular; (4) an elongated quadrate process of the pterygoid; (5) maxillary teeth implantation thecodont anteriorly, aulacodont posteriorly; (6) large body size. On the basis of our observations (detailed in the bone-by-bone descriptions below), specimen GPIT 1793/1 shares all these characters with the holotype of B. leptorhynchus (except character 4, not visible in the latter) and with all other specimens examined (BES SC 1016, PIMUZ T 4376, PIMUZ T 4847 and PIMUZ T 1895), wherever the characters are preserved (e.g., the pterygoid character is shared by GPIT 1793/1, PIMUZ T 4847 and BES SC 1016).

In general, no autapomorphies supporting and justifying a distinction of Mikadocephalus gracilirostris from Besanosaurus leptorhynchus can be found in GPIT 1793/1 and BES SC 999. For these reasons we consider Mikadocephalus gracilirostris a junior synonym of Besanosaurus leptorhynchus, as previously hypothesized by Sander (2000: 15) and McGowan & Motani (2003: 127). Consequently, subsequent assignment of the PIMUZ shastasaurid material to Mikadocephalus gracilirostris (i.e., PIMUZ T 4376; Maisch & Matzke, 2000) must be rejected, and the valid taxon name for all the referred material remains by priority Besanosaurus leptorhynchus Dal Sasso & Pinna, 1996, according to the ICZN (1999).

Maisch & Matzke (2000) deemed the relative size of the skulls of Cymbospondylus and Besanosaurus as very small when compared to body length, equaling about “one-quarter of the presacral length”; the authors then considered this character to be in stark contrast to the condition in Mikadocephalus gracilirostris, whose skull, referring to PIMUZ T 4376 was “more than half the length of the presacral vertebral column”. This has been regarded by Maisch & Matzke (2000) as a valid diagnostic character, reported as clearly unrelated to ontogeny and distinguishing Besanosaurus from Mikadocephalus, and Mikadocephalus from other shastasaurids known at that time. However, given that in most ichthyosaurs the cranium displays negative allometry vs body size during growth (McGowan, 1973), we deem it plausible that this difference can be attributed to ontogeny, and therefore we consider the six specimens described as a possible ontogenetic series of B. leptorhynchus (discussed below).

Moreover, our analysis shows that there are no observable qualitative characters of the skull and mandible that differ between the proposed holotypes: treating the proposed taxa as synonymous is therefore logical. In addition, Maisch & Matzke (2000) compared two different (i.e., non-homologous) lengths (presacral vertebral column vs presacral length) to highlight that anatomical difference between Besanosaurus and Mikadocephalus as diagnostic: it is true that the skull of PIMUZ T 4376 is a few centimeters longer than half the length of the presacral vertebral column, but is also true that the holotypic skull of Besanosaurus is around one-third of the presacral length of the vertebral column (which equals, as the authors stated, “one-quarter of the presacral length”, a length that includes the skull itself and not just the presacral vertebral column). This different ratio can be explained by intraspecific (possibly ontogenetic) variation, as we demonstrate below.

Maisch & Matzke (1997a) also mentioned that the interpterygoid vacuity of Mikadocephalus must have been large, and similar to post-Triassic ichthyosaurs. In reality, this cannot be unambiguously determined since a good portion of the medial border of the left pterygoid in GPIT 1793/1 has been broken and dislocated above the rest of the bone. On the other hand, in BES SC 1016, where the two pterygoids are still semi-articulated, the interpterygoid vacuity is narrower than that hypothesized for the holotype of Mikadocephalus. This space is actually larger than in Mixosaurus, showing an intermediate condition between Lower Triassic ichthyosaurs and post-Triassic taxa. It is also comparable in size and morphology to other shastasaurids (Guizhouichthyosaurus tangae, IVPP V11853; ‘Callawayia’ wolonggangense, SPCV 10305; personal observation).

Finally, Maisch & Matzke (1997a) considered the ‘triangular process’ of the quadrate (Maisch & Matzke, 1997a: fig. 7) as a diagnostic character of Mikadocephalus gracilirostris. Interestingly, the right quadrate of a referred specimen of Guanlingsaurus liangae (SPCV 03107; personal observation) also shows such a ‘triangular process’ (Fig. S4). Furthermore, the B. leptorhynchus holotype (BES SC 999), and referred specimens PIMUZ T 4376 and T4847, also possess this character (see quadrate description). Therefore, we consider it more plausible to provisionally treat this character as a possible shastasaurid synapomorphy, rather than a Besanosaurus apomorphy (presence/absence of ‘triangular process’ not determined for other shastasaurids except Besanosaurus and Guanlingsaurus).

Given that our comparison with the former holotype of M. gracilirostris found a substantially identical osteology, both in the shape and interrelationships of the bones, we propose the junior synonymy of the taxon with respect to B. leptorhynchus. As a consequence, the former holotype and only specimen of M. gracilirostris is here redescribed and discussed together with all other Besanosaurus material.

Systematic palaeontology

ICHTHYOPTERYGIA Owen, 1840

ICHTHYOSAURIA Blainville, 1835

MERRIAMOSAURIA Motani, 1999

BESANOSAURUS Dal Sasso & Pinna, 1996

Besanosaurus leptorhynchus Dal Sasso & Pinna, 1996

Type and only species. Besanosaurus leptorhynchus Dal Sasso & Pinna, 1996; middle Besano Formation (uppermost Anisian, Middle Triassic), Monte San Giorgio, Italy/Switzerland.

Type specimen. Complete semi-articulated skeleton, labelled as BES SC 999 in the catalogue of the MSNM (BES SC is an acronym for Besano Sasso Caldo), and coded as 20.S288-2.2 in the Inventario Patrimoniale dello Stato (State Heritage Database).

Type locality. Sasso Caldo site, Besano, Varese Province, NW Lombardy, N. Italy. Geographical coordinates: 45°54′03.7″N 8°55′10.6″E, elevation 650 m.

Type horizon and distribution. middle Besano Formation (sensu Bindellini et al., 2019), uppermost Anisian (N. secedensis Zone sensu Brack et al., 2005), Middle Triassic.

Referred material. PIMUZ T 4376 (complete semi-articulated skeleton with the best-preserved skull of the taxon), PIMUZ T 1895 (incomplete semi-articulated skeleton with well-preserved skull), PIMUZ T 4847 (incomplete semi-disarticulated skeleton with disarticulated skull), GPIT 1793/1 (disarticulated skull, formerly the holotype of Mikadocephalus gracilirostris, preserved in three slabs now re-labelled GPIT-PV-76245 (snout), GPIT-PV-76246 (skull roof), and GPIT-PV-76247 (jugal and lower jaw)), BES SC 1016 (incomplete semi-disarticulated skull; the specimen is coded as 20.S288-2.6 in the Inventario Patrimoniale dello Stato—State Heritage Database).

Emended cranial diagnosis. Large ichthyosaur with one possible autapomorphy—a caudoventral exposure of the postorbital in the temporal region—and the following combination of cranial characters: extremely long, slender, and gracile snout; frontal rostrocaudally elongate and relatively flat; frontal participation in the temporal fossa but not to the temporal fenestra; L-shaped jugal; ‘triangular process’ on the medioventral border of the quadrate; prominent coronoid (preglenoid) process of the surangular, distinctly rising above the dorsal margin of the surangular; tiny conical teeth with a coarsely-striated crown surface and deeply striated roots; mesial maxillary teeth set in sockets; distal maxillary teeth set in a groove shorter than half of the rostral ramus of the maxilla.

RESULTS: REVISED CRANIAL OSTEOLOGY OF BESANOSAURUS LEPTORHYNCHUS

Cranial openings

External naris. It is difficult to infer the shape of the external naris on the holotype of B. leptorhynchus BES SC 999 (Fig. 4). In GPIT 1793/1 the naris is not clearly preserved since the rostrum is disarticulated and the caudal ends of the premaxillae are likely incomplete. In PIMUZ T 4376, although the skull is in very good condition, the rostral portion of the right naris is damaged and the elements bordering it are severely crushed. The best-preserved naris can be seen in specimen PIMUZ T 1895 (Fig. 11): it is long and slender. The dorsal margin of the naris is built by the caudodorsal process of the premaxilla and by a short portion of the lateroventral margin of the nasal, whereas the ventral margin is made by the premaxilla, rostrally, and the maxilla, mediocaudally. The external naris is pointed into a sharp apex rostrally and its rostrodorsal and rostroventral margins are formed by two short and sharp caudal processes of the premaxilla (supranarial and subnarial process, the latter being shorter in length); the caudal margin of the external naris is bordered by the nasal (caudodorsal half) and by the postnarial process of the maxilla (caudoventral half). The caudal portion of the external naris is dorsoventrally wider. In PIMUZ T 1895 and BES SC 1016 the rostral margin of the left naris is more intact than in all other specimens, showing that the supranarial process of the premaxilla covers approximately 50% of the dorsal margin of the external naris, whereas the subnarial process likely covers one-third of the ventral margin of the external naris. The processes are not well-preserved in PIMUZ T 4376, but in GPIT 1793/1 and BES SC 999 (Fig. 12) the subnarial process is partially preserved and similar in length to PIMUZ T 1895 and BES SC 1016, whereas the supranarial processes are broken off. Therefore, very likely all specimens originally possessed the same morphology as that seen in PIMUZ T 1895 and BES SC 1016.

Figure 11 External naris of Besanosaurus leptorhynchus.

External naris and perinarial region of Besanosaurus leptorhynchus. (A) PIMUZ T 1895; (B) BES SC 1016. The white arrows point to the rostral tip of the external naris in both specimens. Scale bars represent 5 cm .

Figure 12 Maxillae of Besanosaurus leptorhynchus.

Maxillae referable to Besanosaurus leptorhynchus. (A and A′) BES SC 999; (B and B′) GPIT 1793/1; (C and C′) PIMUZ T 4376. Light gray areas highlight the maxilla, darker gray areas pinpoint the external naris, when preserved. Abbreviations: see text. Scale bars represent 2.5 cm.

In well-preserved skulls of Guizhouichthyosaurus tangae (IVPP V11869, personal observation; Maisch et al., 2006b; Fig. S1) and in the type specimen of ‘Callawayia’ wolonggangense (SPCV 10306; personal observation) the external naris is bordered by the premaxilla rostrodorsally and rostroventrally, the nasal caudodorsally, and the maxilla caudoventrally: the premaxilla bifurcates caudally into a prominent supranarial process and a smaller subnarial process, thus contributing also to the rostroventral margin of the external naris, with the maxilla not forming the entire ventral margin. In Shonisaurus sp. (Callaway & Massare, 1989: fig. 3; Motani, 1999) the anatomy is very similar, except that the nasal is excluded from the caudodorsal margin of the external naris by the premaxilla. In all three taxa, as in Besanosaurus , the external naris is proportionally long rostrocaudally and narrow dorsoventrally, especially in its rostral portion. The circumnarial region of Besanosaurus is more similar to Guizhouichthyosaurus and ‘Callawayia’ wolonggangense, as a nasal contribution is visible caudodorsally, and the ventral margin is formed by the premaxilla and the maxilla. In Guizhouichthyosaurus (IVPP V11869; Fig. S1) the supranarial process is more dorsoventrally expanded than in Besanosaurus and rostrocaudally reaches approximately 50% of the length of the external naris, whereas the subnarial process is slightly shorter and smaller than the supranarial process; in ‘Callawayia’ wolonggangense (SPCV 10306; personal observation) the supranarial process is stouter than in Besanosaurus, extending at least up to approximately 50% of the dorsal length of the external naris, and the subnarial processes almost reaches the caudal margin of the external naris.

Orbit. The orbit is the largest cranial opening. In Besanosaurus it is oval-shaped, as tall as the scleral ring and 4/3 rostrocaudally longer than high. It is bordered by the jugal ventrally, the postorbital caudally, the postfrontal dorsally, the prefrontal rostrodorsally, and the lacrimal rostrally. The best-preserved orbit can be observed in specimen PIMUZ T 4376, where almost all the bones are still articulated, or semi-articulated (jugal, lacrimal). In PIMUZ T 1895 the contacts of the elements are the same, and in all other specimens that we refer to B. leptorhynchus the disarticulated circumorbital elements show very similar shapes and proportions. The anatomy of the orbit of Besanosaurus is more similar to that in Guizhouichthyosaurus (IVPP V11869; Maisch et al., 2006b) than to ‘Callawayia’ wolonggangense (SPCV 10306; Chen, Cheng & Sander, 2007), in which all the bones that border the orbit seem to be reduced to accommodate a bigger orbit.

Supratemporal fossa. The supratemporal fossa, that includes the supratemporal fenestra, is built by a portion of the parietal medially, of the frontal rostromedially, of the postfrontal rostrolaterally, of the postorbital laterally, and of the supratemporal laterocaudally and caudomedially. In Besanosaurus the supratemporal fossa includes a wide semioval rostral terrace (‘anterior terrace’ of Motani, 1999: char. 14), almost as long as the supratemporal fenestra. This terrace is composed of the caudal two-fifths of the frontal, a medial flange of the postfrontal, and in minor part by a rostrolateral process of the parietal. The supratemporal terrace is nicely preserved almost in 3D in specimen PIMUZ T 4376, well-exposed in PIMUZ T 1895, but not visible in the rotated skull roof of BES SC 999 and GPIT 1793/1. The supratemporal fossa seems to be proportionally smaller in Besanosaurus than in Guizhouichthyosaurus tangae (IVPP V11865; personal observation) and Guanlingsaurus liangae (SPCV03107; personal observation); on the other hand, Besanosaurus ‘anterior terrace’ is bigger than in Shastasaurus (UCMP 9017; Sander et al., 2011) and proportionally similar in size to that in ‘Callawayia’ wolonggangense (SPCV 10306; personal observation).

Supratemporal fenestra. This fenestration, seen in dorsal view, occupies half the area of the supratemporal fossa and opens on its caudolateral border. The best articulated specimens (PIMUZ T 4376 and PIMUZ T 1895) clearly show that it is bordered by the parietal medially, the postfrontal rostrally, the postorbital laterally, and the supratemporal laterocaudally. The frontal does not directly build the profile of the temporal fenestra, although it indeed builds a great portion of the ‘anterior terrace’ of the supratemporal fossa. The same can be observed in GPIT 1793/1 and BES SC 999, although the fenestration is exposed in ventral (internal) view. The lateral margin of the supratemporal fenestra coincides with that of the supratemporal fossa. The supratemporal fenestra of Besanosaurus is one of the smallest observed in the clade Shastasauridae, comparable to that in ‘Callawayia’ wolonggangense (SPCV 10305; Chen, Cheng & Sander, 2007). As in Guizhouichthyosaurus (Maisch et al., 2006b), but unlike ‘Callawayia’ wolonggangense (Chen, Cheng & Sander, 2007), and Guanlingsaurus liangae (Sander et al., 2011), the frontal does not contribute to the rostral border of the fenestra.

Parietal foramen. In Besanosaurus the parietal foramen is located between the two parietals and the two frontals. It opens along the medial sagittal line at about the level of the caudal margin of the scleral rings, and its rostral border is at the level of the rostral border of the supratemporal fenestrae. This is best seen in PIMUZ T 1895 and PIMUZ T 4376, but a small elongate parietal foramen, positioned across the parietals and the frontals, is also preserved in BES SC 999, and in GPIT 1793/1. In PIMUZ T 4376 the parietal foramen is rostrally bordered by the caudal-most ends of the frontals and is positioned mainly between the rostromedial processes of the parietals, which do not contact each other on the midline. This condition differs from the one observed in Guanlingsaurus liangae (Sander et al., 2011), where the parietal foramen is placed more caudally and it is fully enclosed in between the two rostromedial processes of the parietal. For this reason, the parietal foramen of Besanosaurus is more similar to that of Guizhouichthyosaurus tangae and ‘Callawayia’ wolonggangense, although it is proportionally slightly smaller than in the latter taxon. In contrast, in some Parvipelvia, the frontals form most of the rostral and lateral margins of the parietal foramen, only leaving the caudal-most margin to the parietal (McGowan, 1973; Kear, 2005).

Foramen magnum. The preservation of the foramen magnum is exceedingly rare in ichthyosaurs due to the nature of the ossification of the braincase; in fact, it is not preserved in its integrity in any of the examined fossils of Besanosaurus. Nevertheless, its profile can be inferred from two specimens: PIMUZ T 4376, where the basioccipital shows its dorsal surface and very well-preserved facets for the exoccipitals; and GPIT 1793/1, where the supraoccipital forms the dorsal margin of the foramen magnum. In caudal view, it should possess a sub-pentagonal profile, with rounded margins and equal sides, possibly being slightly taller than wide. These margins are defined by four different bones, which in clockwise order are the supraoccipital, the right exoccipital, the basioccipital, and the left exoccipital.

Scleral plates

The sclerotic ring is complete and well-preserved in PIMUZ T 4376; it is also almost complete, although partially disarticulated, in PIMUZ T 1895. At least seven distinct scleral plates, of which six are in articulation (and somewhat imbricate), are still positioned in the left orbit of BES SC 999, only partially covered by a fragment of the putative left palatine, and slightly displaced rostrally with respect to their original position inside the orbit. These plates form almost one half of the entire sclerotic ring that occupied a major part of the orbit. On the whole, we can thus calculate that the sclerotic ring of Besanosaurus was composed of approximately 15–17 plates.

Each scleral plate is subrectangular to trapezoidal and characterised by fine, sometimes undulating striations that radiate from the base to the lateral surface. Here the plates bend medially at about 45°, showing a short peripheral surface measuring about 1/3 of the lateral surface.

Compared to Parvipelvia, the sclerotic ring of the Besanosaurus holotype is proportionally smaller if normalised to the body length of the animal (see Motani, Rothschild & Wahl, 1999 for the discussion of this ratio), albeit being bigger than that in Cymbospondylus (PIMUZ T 4351; Sander, 1989) and similar in size to Mixosaurus (BES SC 1000; Renesto et al., 2020). The number of sclerotic plates in Besanosaurus is also similar to that in Mixosaurus (McGowan & Motani, 2003: 24).

Dermal skull roof

Premaxilla. The holotype of B. leptorhynchus (BES SC 999) preserves complete and articulated rostral elements. The slender rostrum which inspired its specific name entirely comprises narrow and elongate dentary and premaxillary bones. The latter likely overlap for some extent the lateral surface of the maxillae, and caudomedially accommodate equally narrow and very elongate vomers, as shown by the disarticulated vomers exposed in specimen GPIT 1793/1, and by the articular surfaces of the almost identical premaxillae. Complete premaxillae are also preserved in PIMUZ T 4376, where both the left and right elements are visible.

In both BES SC 999 and PIMUZ T 4376, the premaxilla rostrally terminates in a rounded tip, which extends more rostrally than the tip of the dentary, thus resulting in an overbite of less than one cm (some postmortem deformation might have occurred). The tips of the premaxillae are missing, like most of the rostral halves of the jaws, in BES SC 1016, which in turn helps to describe the premaxillary-maxillary contacts.

In the holotype of B. leptorhynchus, the ventral surface of the premaxilla hosts several deep sockets that are also clearly visible in BES SC 1016 (Fig. 10). The premaxilla of BES SC 999 hosts at least 35 (estimated) teeth that interlock with the dentary teeth.

The left premaxilla in this specimen still articulates caudoventrally with the left maxilla and dorsomedially with a long and slender rostral process of the left nasal. The premaxilla terminates caudally with very thin supranarial and subnarial processes (better described above).

In the holotype BES SC 999, a caudorostrally extended area of collapsed premaxillary bone is visible in front of the naris; this possibly highlights the presence of an internal weak point. Something similar can also be observed in PIMUZ T 4376 and T 1895 and, curiously, in the same area, the right premaxilla of GPIT 1793/1 is also collapsed and fractured along a horizontal line that was described by Maisch & Matzke (1997a) as the anterior (i.e., rostral) margin of the external naris. This, actually, could represent the rostral-most extension of a possible pneumatic recess of the internal naris, or a neurovascular pathway. The subnarial and supranarial processes, described by Maisch & Matzke (1997a) for GPIT 1793/1 are here considered taphonomic artefacts. If the relative lengths of the supranarial and subnarial processes in PIMUZ T 4376 were similar to those reconstructed by Maisch & Matzke (1997a), the external naris of the specimen would be unnaturally long, proportionally even longer than in the reconstruction published by Maisch & Matzke (2000).

Maxilla. The maxilla of Besanosaurus (Fig. 12) is a craniocaudally elongate triradiate bone, as in the majority of the other Triassic ichthyosaurs (e.g., Ji et al., 2016: figs. 3 and 4). It is well-exposed in most of the specimens examined here, with some variation in the shape and height of the postnarial process, that we interpret as intraspecific or taphonomic variation (it is more pronounced in GPIT 1793/1 and BES SC 1016). Similarly, in BES SC 999 the main body of the maxilla looks dorsoventrally taller than in the other specimens, due to taphonomic mediolateral compression. In the holotype, the long and slender rostral ramus of the maxilla bears around 20 (estimated) teeth that are present along the dentigerous margin from the rostral tip of the bone to the level of the rostral border of the naris. The caudal-most, tooth-bearing portion of the maxilla, below the external naris hosts the teeth in a short groove (aulacodont implantation), although 70% of the maxillary teeth are set in distinct alveoli (subthecodont implantation) well visible in GPIT 1793/1. The caudodorsally directed and curved postnarial process delimits the caudal border of the external naris and accommodates the convex rostral margin of the lacrimal. This process projects toward the prefrontal forming a contact point (PIMUZ T 4376), excluding the lacrimal from contact with the nasal. Similarly, the postnarial process forms a point contact with the prefrontal in Guizhouichthyosaurus tangae (IVPV 11869; personal observation) and ‘Callawayia’ wolonggangense (SPCV 10306, personal observation), a condition contrasting with Guanlingsaurus liangae where this process is short and well separated from the prefrontal (SPCV 03107, personal observation; Sander et al., 2011). The caudal ramus of the maxilla, which measures three-fifths the length of the rostral ramus, is toothless and in BES SC 999, PIMUZ T 4376, and T 1895 slightly bends caudally below the lacrimal, also following the natural bend of the dentary and surangular below the orbit. In GPIT 1793/1, as reported by Maisch & Matzke (1997a), the caudal rami of both maxillae are incomplete and are therefore shorter than in the Besanosaurus holotype, and toothless as well.

Specimens GPIT 1793/1, PIMUZ T 1895 and PIMUZ T 4376 clearly show two maxillary foramina below the naris, which extend rostrally into shallow, rostrocaudally elongate grooves (best preserved in GPIT 1793/1). Similar foramina are present in Guizhouichthyosaurus tangae (IVPP V11869, personal observation; GNG dq-46, Maisch et al., 2006b; Fig. S1) below the postnarial process of the maxilla. Neurovascular foramina, albeit larger and more rounded in outline, are also present in the maxilla of Guanlingsaurus liangae (SPCV 03107, personal observation).

On the whole, the maxilla of Besanosaurus maintains similar proportions to those observed in ‘Callawayia’ wolonggangense (SPCV 10306, personal observation), being relatively thin and slender, clearly less robust than in Shonisaurus popularis and Cymbospondylus (Camp, 1980; Sander et al., 2011). The caudal ramus of the maxilla slightly bends ventrally as in Shastasaurus pacificus ( Sander et al., 2011). However, the postnarial process is absent in Shonisaurus popularis (Camp, 1980; Callaway & Massare, 1989).

Nasal. The left and right nasal bones are best preserved in PIMUZ T 4376 and GPIT 1793/1, exposed in dorsal and ventral view, respectively; in BES SC 999 they are severely deformed and fractured. The nasals of Besanosaurus are very slender elongate elements. They are triangular in shape and taper extremely towards the rostral region in between the premaxillae. The caudal-most portion of the bone is broader than the rostral, reaching its maximum mediolateral width at the straight lateral margin that contacts the dorsal edge of the postnarial process of the maxilla (PIMUZ T 4376). Rostrally to that edge, the nasal shows a small facet where a portion of the dorsal tip of the postnarial process of the maxilla is housed (GPIT 1793/1 and PIMUZ T 1895). The nasal borders the caudalmost dorsal edge of the naris and contributes to the dorsal side of the rostrum with a very long premaxillary contact. The nasal contacts the prefrontal laterally, but does not contact the postfrontal. Dorsocaudally the nasal forms an interdigitating suture with the frontal.

In dorsal view, the surface of the best-preserved nasal (PIMUZ T 4376) appears convex and shows sub-longitudinal striations. In ventral (i.e., internal) view (GPIT 1793/1) the nasal shows a concave smooth surface for most of its length, and a flat rugose texture in the area of the suture with the frontal; a medial dorsocaudally elongate and thin vertical process might represent the contact with the vomer.

Interestingly, the nasals of BES SC 999 and PIMUZ T 4376 are fractured where they begin to narrow rostrally, becoming thinner and slender; similar fractures are also observed in PIMUZ T 1895. This feature highlights the presence of a distinct constriction separating the gracile rostrum from the rest of the skull, as seen in BES SC 999 and PIMUZ T 4376. Just rostrally to this constriction, at least four neurovascular foramina are present in PIMUZ T 1895; similar foramina are also visible in Guizhouichthyosaurus tangae (IVPP V11869, personal observation, Fig. S1; GNG dq-46, Maisch et al., 2006b).

Lacrimal. The lacrimal of Besanosaurus leptorhynchus is approximately comma-shaped in outline in lateral view. This element displays a convex rostral margin, a concave caudal edge, and a caudoventrally pointed tip. This morphology is best seen in the right lacrimal of PIMUZ T 4376 and the left lacrimal of T 1895, which also preserve them in articulation with all other elements: the dorsal margin contacts the rostroventral process of the prefrontal; the convex rostral margin articulates with the postnarial process and a portion of the caudal ramus of the maxilla; and the caudoventral tip contacts the rostral tip of the jugal, ventrally. The concave caudal edge marks the rostral limit of the orbit.

In GPIT 1793/1 the lacrimal bones are semi-articulated and exposed in medial (internal) view, showing a concave surface and a ridged margin arising towards the pointed tips, which represents the contact with the caudal ramus of the maxilla. In BES SC 999, the putative (and deformed) left lacrimal does not show a clear articulation with the other bones.

The lacrimal of Besanosaurus is similar to Guizhouichthyosaurus tangae (GNG dq-46; Maisch et al., 2006b) and ‘Callawayia’ wolonggangense, which namely shows a less robust anatomy (more elongate overall shape, more slender, curved and pointed tips) linked to the presence of a larger orbit (SPCV 10306, personal observation; Chen, Cheng & Sander, 2007). Platypterygius australis shows a lacrimal semilunate shape (Kear, 2005), somewhat similar to that in Besanosaurus, although its caudoventral extension is less developed.

Prefrontal. Although in the holotype of B. leptorhynchus the anatomy of the prefrontal is unclear, this bone is well-preserved in three other specimens (PIMUZ T 1895, T 4376, and GPIT 1793/1). The prefrontal in lateral view is a comma-shaped bone angled by 45°, with a rostroventrally tapered tip contacting the dorsal end of the lacrimal and the caudal end of the maxillary postnarial process. The larger caudal portion contacts the postfrontal caudolaterally, and the frontal mediodorsally. The lateroventral margin of the prefrontal is part of the orbital rim, whereas the rostromedial margin is sutured to the nasal. The lateral portion of the prefrontal is mediolaterally thickened and separated from the medial portion by a rounded ridge, that becomes even more pronounced caudally, at the level of the ‘anterior terrace’ of the temporal fenestra. This thickened area continues along the dorsal border of the orbit towards the postfrontal, building a prominent supraorbital crest. As a result, the cross-section of the whole skull looks roughly hexagonal-shaped, with a rounded, slightly protruding crest at the lateral edges of the prefrontal and the postfrontal.

Postfrontal. The anatomy of the postfrontal is best observed in specimen PIMUZ T 4376, although it is also visible on the left and right sides in PIMUZ T 1895 and GIPT 1793/1 (in ventral view, Fig. 13). The postfrontal occupies a surface of the skull roof immediately caudal to the prefrontal and is contacted by this bone rostrally. Its lateral edge is thick and sigmoid in dorsal view. This area is separated by a clear step from the medial, triangular-shaped, rostromedial process of the postfrontal. Its rostral part is elevated in comparison to its caudal part, which contributes to the rostrolateral half of the ‘anterior terrace’ of the supratemporal fenestra. Medially the rostromedial process contacts the caudal half of the frontal bone, with an irregularly-shaped suture that in ventral view appears deeply interdigitating (GIPT 1793/1). Caudomedially the caudomedial process contacts the rostrolateral ramus of the parietal. The stout pointed caudal ramus of the postfrontal has a minor contribution to the lateral border of the temporal fenestra and contacts the dorsal portion of the postorbital (PIMUZ T 4376). Here, the suture is sharp and oblique, with the postfrontal laterally overlapping part of the postorbital, which possesses a concave facet that houses the caudal end of the postfrontal.

Figure 13 GPIT 1793/1 under UV light.

GPIT 1793/1, partly under visible, partly under UV light (365 nm), highlighting the sutures, otherwise not easily distinguishable from fractures, of frontals and adjacent bones on the ventral (internal) side of the skull roof: the extent of the frontals is greater internally than externally (see Figs. 4–6). Scale bar represents 5 cm.

Unlike in Euichthyosauria, the postfrontal does not exclude the postorbital from bordering the temporal fenestra, a condition which is in contrast with a recent interpretation given for Cymbospondylus duelferi, Cy. petrinus and Cy. nichollsi (Klein et al., 2020).

Postorbital. A purported postorbital in the holotype of B. leptorhynchus (Dal Sasso & Pinna, 1996: fig. 10; Fig. 4) is here regarded as a possible quadratojugal (see below). The only certain postorbital is seen in PIMUZ T 4376, still articulated to adjacent elements of the left temporal region. It produces a somewhat hourglass-shaped lateral surface, an approximately trapezoidal dorsolateral exposure, and a caudoventral exposure which tapers into a distinct apex. Distinct striations radiate from the rostrolateral corner of the bone to the whole surface of this element, distinguishing it from the rugose surfaces of the postfrontal, the supratemporal and the squamosal, with striations parallel to the lateral margin of the temporal fossa. The caudal and the rostral borders of the postorbital are concave, with the latter forming the caudal margin of the orbital rim and the dorsomedial margin contributing to part of the supratemporal fossa and fenestra. In PIMUZ T 4376 the caudal contact with the squamosal is extensive, where its rostral margin seems to overlap the caudal margin of the postorbital.

The bulky aspect of the postorbital of Besanosaurus differs from the thin more “semilunar-shaped” postorbital of more derived Euichthyosauria, being more similar in shape to that of Shastasaurus pacificus (Sander et al., 2011) or Guizhouichthyosaurus tangae (Maisch et al., 2006b; IVPP V11865 and V11869, personal observation; Fig. S1), with which in particular it shares a comparable anatomy, showing a similar profile and the same bone contacts. As in the latter, but with lesser extent, in Besanosaurus the postorbital contributes to the lateral border of the temporal fenestra, preventing the postfrontal-supratemporal contact. In Shastasaurus (Sander et al., 2011), the caudolateral process of postfrontal is longer, but the postorbital still contributes to the lateral margin of the supratemporal fenestra. In ‘Callawayia’ wolonggangense (SPCV 10305, personal observation), and in a single specimen referred to Guizhouichthyosaurus tangae (GNG dq-41, Maisch et al., 2006b), the postorbital is excluded from the lateral margin of the supratemporal fenestra by an extension of the laterocaudal process of the postfrontal contacting the supratemporal.

The base of the postorbital certainly contacts the horizontal ramus of the jugal, but this articulation cannot be seen in any Besanosaurus specimens, because all jugals are more or less displaced from their original position. In PIMUZ T 4376, the postorbital forms a ventrocaudal exposure in the temporal region—a possible autapomorphy of B. leptorhynchus (difficult to compare with other shastasaurids, in which the temporal region is often deformed or incompletely preserved).

Jugal. The jugal is preserved in all specimens, although variably—and sometimes bizarrely (BES SC 999)—disarticulated. It is almost in place in PIMUZ T 4376, in which it is slightly shifted caudodorsally. It is disarticulated, well-visible and exposed in medial view in GPIT 1793/1 (Fig. 8). The jugal is a mediolaterally compressed, L-shaped bone, with a proportionally long horizontal suborbital ramus and a shorter vertical postorbital ramus. The two rami are orientated at an angle of almost 90°, which accommodates the ventrocaudal border of the orbit. In our interpretation, the rostral tip of the jugal—which is triangularly tapering into two oblique facets—is situated between the caudal tip of the lacrimal dorsally and the caudal ramus of the maxilla ventrally; on the other hand, the postorbital (dorsal) ramus enters the postorbital region and is thereby partially hidden by the postorbital and the squamosal contacting them with its smooth lateral surface. Such a loose articulation seems to be a natural condition, with the postorbital ramus of the jugal possibly interposed between the quadratojugal medially and the squamosal laterally. In fact, excluding PIMUZ T 4376, in all the specimens assigned here to B. leptorhynchus, the jugal is disarticulated from the other elements and does not show any apparent suture or scar on the surface of the caudal ramus; even in PIMUZ T 4376 it seems to be slightly displaced dorsally.

The suborbital (ventral) ramus varies quite in cross-sectional shape: it is dorsoventrally expanded rostrally, but it tapers into a smooth and suboval shape in its caudal half. The jugal has a concavity on its medioventral side in the area rostral to the bend of the postorbital ramus. This morphology is exposed in the holotype (BES SC 999), in PIMUZ T 4876, and in GPIT 1793/1. In this medioventral concavity, or groove, the surangular was likely housed while the animal was keeping its jaws closed. Judging from the position of the jugals, this feature can also be seen in Guizhouichthyosaurus tangae (Maisch et al., 2006b: fig. 2) and Utatsusaurus hataii (Motani, Minoura & Ando, 1998: fig. 2).

Squamosal. The shape of the squamosal is clearly visible in specimen PIMUZ T 4376 and partly visible in PIMUZ T 1895, whereas in the holotype BES SC 999 this element seems to be present below the slab surface, visible only through CT scans. It is an approximately quadrangular-shaped, mediolaterally flat bone, proportionally bigger than the quadratojugal (with a greater external exposure than the quadratojugal), and dorsoventrally shorter than the postorbital. Dorsally it contacts the supratemporal and rostrally the postorbital, whereby it overlaps its caudal margin. On its medial side, the squamosal contacts the quadratojugal, and rostrally these two bones are separated by the postorbital ramus of the jugal (Figs. 4–6), as seen in Guizhouichthyosaurus tangae (IVPP V11865, personal observation; Fig. S2). Several fine striations project dorsoventrally from the dorsocaudal edge of this element. The ventral margin of the squamosal bears two short processes: the caudal process is slightly longer and is directed caudoventrally, the rostral process is shorter and directed ventrally. The cranioventral edge of the quadratojugal is visible laterally below the processes. The large size of the squamosal and its prominence in the cheek region is a common feature of Shastasauridae as seen in Guanlingsaurus liangae (Sander et al., 2011), Guizhouichthyosaurus tangae (IVPPV 11869, personal observation; Maisch et al., 2006b), Shastasaurus pacificus (Sander et al., 2011) and possibly Shonisaurus (Callaway & Massare, 1989). However, the squamosal seems to be more quadrangular in Besanosaurus than in these taxa. This element becomes reduced in size throughout ichthyosaur phylogeny. As the cheek region becomes less prominent in Parvipelvia, the squamosal adopts a more triangular morphology as seen in e.g., Ichthyosaurus and Hauffiopteryx (e.g., Marek et al., 2015). Similarly, the squamosal seems to be more triangular in ‘Callawayia’ wolonggangense, but its caudal margins are broken in SPCV 10306, which might affect the perceived shape (personal observation).

Frontal. The frontal bones are preserved, co-articulated, and exposed in dorsal view in PIMUZ T 4376 and PIMUZ T 1895, and in ventral view in the holotype BES SC 999 and in GPIT 1793/1. A pair of putative frontals are also visible in PIMUZ T 4847 (albeit these fragments may represent only a portion of the ‘anterior terrace’ of the temporal fenestra since they show the same radiate bone texture as on the dorsal surface of the skull roof in PIMUZ T 4376 and PIMUZ T 1895). In Besanosaurus the frontal contacts the nasal rostrally, the parietal caudally, the prefrontal rostrolaterally, and the postfrontal caudolaterally. The frontal also borders the very rostral edge of the parietal foramen. The frontals contact each other at the midline; this contact is characterized by the presence of a prominent sagittal crest, which starts rostrally to the parietal foramen and becomes less prominent approaching the nasals. In the most 3D-preserved specimen (PIMUZ T 4376), the sagittal crest of the frontals is the continuation of the more prominent sagittal crest of the parietals, the two being separated only by the opening of the parietal foramen.

In ventral view the frontal bone is slightly longer than in dorsal view, where it expands mediolaterally: due to oblique sutural contacts, the prefrontal and the postfrontal overlap the dorsolateral surface of the frontal. Rostrodorsally there is a similar slight overlapping with the nasal, whereas caudodorsally the frontal is overlapped by the rostromedial and rostrolateral rami of the parietal. In GPIT 1793/1, on the ventral side of the frontals, there are remnants of the putative olfactory lobes, partially hidden by broken and ventrally dislocated rostromedial portions of the frontals (Fig. 13). These lobes define a clear step between the rostral half and the caudal half of the frontals; for this reason, the caudal halves are thicker dorsoventrally. As in Besanosaurus, the frontal contributes to the supratemporal fossa in Guizhouichthyosaurus tangae (Maisch et al., 2006b), in ‘Callawayia’ wolonggangense (Chen, Cheng & Sander, 2007), Guanlingsaurus liangae (Sander et al., 2011) and Shastasaurus pacificus (Sander et al., 2011).

Parietal. Similarly to the frontals, the parietals are preserved articulated in dorsal view in specimens PIMUZ T 4376 and T 1895, and in ventral view in GPIT 1793/1 and BES SC 999 (although are quite damaged in the latter specimen). The parietal bone contacts the frontal rostrally, the postfrontal rostrolaterally, the supratemporal caudolaterally, and the supraoccipital caudally. Medially the two parietals contact each other and form a prominent sagittal crest (only visible in PIMUZ T 4376). The two rostral rami of the parietal slightly overlap the caudal edge of the frontal, so that on the ventral side of the skull they seem rostrocaudally longer than on the dorsal side.

The parietal of Besanosaurus is a highly three-dimensional element of the skull roof. The lateral side of the parietal possesses a smooth, concave, and almost vertical wall, which borders the temporal fenestra medially and where the rostral portion of the musculus adductor mandibulae externus profundus (mAMEP) and the ventral half of the musculus pseudotemporalis superficialis (mPSTs) may have attached (as reported in Sphenodon, e.g., Jones et al., 2009). In PIMUZ T 4376 this wall is separated from the prominent sagittal crest by a narrow and short flat surface, in continuity with the ‘anterior terrace’ of the temporal fenestra. The rostral portion of the musculus adductor mandibulae externus medialis (mAMEM) and the dorsal half of the mPSTs may have attached on this flat surface and on the sagittal crest (Jones et al., 2009). Compared to ‘Callawayia’ wolonggangense (SPCV 10306, personal observation) and Guanlingsaurus liangae (SPCV 03107, personal observation; Fig. S4), the sagittal crest of B. leptorhynchus seems more prominent and more similar to that of Guizhouichthyosaurus tangae (IVPP V11865, personal observation), albeit not as prominent as in mixosaurids (e.g., MSNM V 455, personal observation) and Cymbospondylus (PIMUZ T 4351).

Remarkably, in GPIT 1793/1 the parietals exposed in ventral (internal) view display a deep ovoid-shaped concavity that would have accommodated the rostral region of the cerebral hemispheres, possibly including or even exclusively including the optic lobes (Marek et al., 2015; Fig. 13), apparently even more prominent than what is preserved in the Callawayia neoscapularis holotype (ROM 41993; McGowan, 1994). There is no apparent epipterygoid facet on the ventral side of the parietal. Distinct epipterygoid facets are observed in Ichthyosaurus and Platypterygius australis as ventrally directed extensions lateral to the indentation of the rostral region of the cerebral hemisphere (McGowan, 1973; Kear, 2005). It is possible this facet is ossified only in parvipelvian ichthyosaurs. This leads to the hypothesis that there was no (ossified) connection between the epipterygoid and skull roof in Besanosaurus.

Supratemporal. The supratemporal, clearly visible in PIMUZ T 4376, is a triradiate complex bone. The rostrolateral ramus develops on the dorsolateral side of the skull roof and contacts the postorbital rostroventrally, and the squamosal ventrally. A process in the form of a descending ramus functions as a major part of the caudolateral side of the skull and envelops the dorsal head of the quadrate caudally, as seen in Guanlingsaurus liangae (SPCV 03107, personal observation; Fig. S3). The rostromedial ramus is directed rostromedially and contacts, on its rostromedial side, the caudolateral process of the parietal. On the caudal side a facet for the opisthotic is not visible. We interpret the “supratemporal antero-medial extension” (Motani et al., 2017: char. 12) as short on the basis of specimens PIMUZ T 4376 and BES SC 999, as their morphology is short compared with Cymbospondylus (Motani, 1999). This morphological difference has been used as a character in recent phylogenetic analyses (Ji et al., 2016; Huang et al., 2019).

In BES SC 999, both supratemporals show their internal (rostromedial) aspect, characterized by a rugose texture. An apparent L-shaped concavity is also present, in which the dorsal portion of the quadrate head was likely housed; this concavity is dorsally demarcated by two conical prominences, connected by a ridge, and laterally by a longer ridge, descending from the lateral-most prominence. The putative left supratemporal in GPIT 1793/1 is also preserved in rostromedial view: the surface is concave and two prominences, closely resembling the one described above, are visible as well.

Braincase

Supraoccipital. This bone is clearly preserved only in GPIT 1793/1, exposed in caudal view. The supraoccipital is represented by an inverted U-shaped “massively built arched bone” (Maisch & Matzke, 1997a). The inner ventral arch of this bone delineates the upper margin of the foramen magnum, whereas the outer dorsal arch (Fig. 14A) contacted the caudoventral margins of the parietals. Lateral to the foramen magnum, the ventral edges of the supraoccipital arch contain small facets for the exoccipitals. Curiously, the left dorsolateral margin of the supraoccipital hosts a notch that seems absent on the right side of the bone, likely due to oblique compression. This was also noted by Maisch & Matzke (1997a), who suspected this notch to be a neurovascular foramen similar to those described in Ophthalmosaurus icenicus and Ichthyosaurus (Andrews, 1910; McGowan, 1973). This morphology is most similar to Stenopterygius, as that taxon also lacks distinct supraoccipital foramina in most observed specimens, but contrasts with the morphology of most other Euichthyosauria in which the supraoccipital is known, as many taxa have two distinct foramina on the lateral sides of the element (Miedema & Maxwell, 2019).

Figure 14 Braincase elements of Besanosaurus leptorhynchus.

Disarticulated braincase elements of Besanosaurus leptorhynchus GPIT 1793/1 (A–C, G), PIMUZ T 4376 (D and E), and BES SC 999 (F). Interpretative drawings are denoted by apostrophes. (A and A′) supraoccipital in posterior view; (B and B′) opisthotic in rostromedial view. (C and C′) prootic in caudal view. (D and D′) exoccipital in rostral? view; (E and E′) basioccipital in dorsal view; (F and F′) basioccipital in caudal view; (G and G′) parabasisphenoid in ventral view. Dashed lines indicate portions of bones not visible on the surface, thin lines indicate bone structures. Abbreviations: see text. Scale bars represent 1 cm.

Exoccipital. Only one clear exoccipital is present, close to its original position, in specimen PIMUZ T 4376 (Fig. 14D). A putative right exoccipital is partially exposed in rostral view in BES SC 999. This bone in Besanosaurus appears slightly dorsoventrally taller than rostrocaudally long and shaped as a column, with a constricted central body and wider ends. It possesses a ventral facet for the basioccipital and a dorsal facet for the supraoccipital. The latter is smaller, occupying an area that is two-thirds the size of the one occupied by the ventral facet. There is one small indentation in its rostral side, which we interpret as the original location of the hypoglossal foramen, although the opening is now closed due to sediment infill and other diagenetic processes. One hypoglossal foramen is usually present in the exoccipital of Ichthyosauria, although there can be more, as seen in e.g., Temnodontosaurus and Ophthalmosaurus icenicus (Maisch, 1997, 2002).

Prootic. A possible prootic is exposed as a small isolated element in the occipital region of PIMUZ T 1895 and GPIT 1793/1 (Fig. 14C). In each of these specimens, this bone has a size comparable to that of a single scleral plate, and the shape of a discoidal capsule. Rearticulated in anatomical position (e.g., McGowan & Motani, 2003), this bony capsule would have been convex rostrally and concave caudally and, on the caudal side, it would have contacted a cartilaginous otic capsule along its external circumference. The capsule is not preserved in any of the specimens we examined. The prootic in specimen GPIT 1793/1 tentatively displays a V-shaped indentation (although heavily flattened) in which the otic capsule would have fit. The degree of prominence of a possible dividing ridge between the major areas of the indentation is unclear due to the compression. The prootic of Besanosaurus is more similar to that of Lower Jurassic ichthyosaurs than to that of Mixosaurus and cymbospondylid ichthyosaurs, which have a more elongated and quadrangular morphology and are larger in comparison to their opisthotic – a morphology which resembles a condition similar to their diapsid ancestors (Maisch & Matzke, 2006; Maisch, Matzke & Brinkmann, 2006a).

Opisthotic. A well-preserved right opisthotic is exposed in rostral (internal) view in GPIT 1793/1 (Fig. 14D), which was not described by Maisch & Matzke (1997a), because it was unprepared at the time. A putative opisthotic is also preserved in BES SC 999. The opisthotic consists of a rectangular to oval medial head with indentations that accommodated the caudal region of the semicircular canals, a stout paroccipital process, and a clear basioccipital facet. The paroccipital process is roughly as wide as it is tall when measured at its widest position. This morphology resembles the condition in more derived Euichthyosauria, in which the paroccipital process is usually stouter (e.g., Stenopterygius; Miedema & Maxwell, 2019) and Temnodontosaurus trigonodon (Maisch, 2002), than in the more basal Mixosaurus cornalianus or Phantomosaurus neubigi (Maisch & Matzke, 2006). In these latter taxa the paroccipital process is longer than wide and tapers distally, which is not the case in the opisthotic associated with GPIT 1793/1 (Maisch, Matzke & Brinkmann, 2006a; Maisch & Matzke, 2006). The paroccipital process heavily resembles the morphology described for Shonisaurus (Camp, 1980). In GPIT 1793/1 the basioccipital facet is distinctly offset from the paroccipital process and medial head due to the curved convex morphology of the ventral margin of the opisthotic. The morphology of the medial head and shape of the indentations for the semicircular canals are thus far not known in any other Triassic ichthyosaur. The indentations are V-shaped as in all other adult ichthyosaurs in which this morphology is known. The angle between the indentation of the horizontal semicircular canal and caudal vertical semicircular canal are roughly similar to those in other ichthyosaurs. There may be a high degree of conservatism in the inner-ear shape of ichthyosaurs, although this needs further study. The shape of the indentation is different from the early adult morphology in Stenopterygius, although it is unsure whether the shape of the inner ear or the shape of the enclosing opisthotic is more affected by ontogeny (Miedema & Maxwell, 2019). The indentation for the caudal vertical semicircular canal does not reach higher than the dorsal margin of the medial head—this leads to a morphology which differs from many post-Triassic taxa in which this indentation creates a second “process” as visible in caudal view (clearly present in Stenopterygius and Platypterygius australis and to some degree in Ichthyosaurus and Temnodontosaurus (McGowan, 1973; Maisch, 2002; Kear, 2005; Miedema & Maxwell, 2019)).

Stapes. The stapes is elongated and slender. Both stapedes are present disarticulated in the holotype of B. leptorhynchus (Fig. 4): the right one is exposed in rostrolateral view, not obscured by other bones, close to its original position but rotated 180° along the horizontal plane; the left one is partly hidden by the postorbital but visible through CT scans. One stapes is also visible in GPIT 1793/1. All stapedes of Besanosaurus are slender and subcylindrical in shape, with expanded ends and constricted shafts. Both in BES SC 999 and GPIT 1793/1 the lateral articulation for the quadrate is wider and cup-shaped, whereas the sigmoidal medial portion tapers towards the basioccipital, as described by Camp (1980: fig. 12) for Shonisaurus. The stapedes of Besanosaurus, unlike those of Merriamosauria, seem to lack a facet for the opisthotic on their dorsomedial end.

Basioccipital. Two specimens expose the basioccipital close to its original position: the holotype BES SC 999 (Fig. 14F) preserves this bone heavily compressed and exposed in caudal view, whereas in PIMUZ T 4376 (Fig. 14E) the basioccipital is preserved in dorsal view. In caudal view (BES SC 999) a wide extracondylar area is visible below the condyle (Fig. 14F). The area is slightly convex ventrally resembling Ichthyosaurus, but not as convex as in, e.g., Phantomosaurus (Maisch & Matzke, 2006). A wide extracondylar area is characteristic of more basal ichthyosaurs, as the extracondylar area is severely reduced in Stenopterygius and virtually absent in Ophthalmosauridae (Miedema & Maxwell, 2019). The convex occipital condyle is located above this area and occupies a medial portion of the basioccipital bone that is half the size of the extracondylar area. In PIMUZ T 4376 (Fig. 14E) the rostrocaudal length of the basioccipital appears close to its transverse length, which is consistent with the typical robust aspect of this element. On the dorsal surface of the bone, the two facets for the exoccipitals are well-defined; these are represented by two shallow drop-shaped depressions, tapered rostrally. The two exoccipital facets are also parallel and very close to each other, as seen in Stenopterygius (Miedema & Maxwell, 2019), and are separated by a dorsal median ridge (i.e., the floor of the foramen magnum). Rostral to the medial ridge a prominent basioccipital peg emerges, laterally delimiting two subtriangular articular facets for the basisphenoid. Through CT scans, the basioccipital peg is also visible in the Besanosaurus holotype (Fig. 10).

Parabasisphenoid. GPIT 1793/1 preserves an almost intact parabasisphenoid exposed in ventral view (Fig. 14D), although it is disarticulated from the rest of the skull and the cultriform process is likely incomplete rostrally. In general, the parabasisphenoid has a piriform profile, being wider caudally than rostrally. In fact, a deep constriction is present in the middle of its dorsoventrally compressed body, just caudal to the articular surface of the basipterygoid process, as also seen in Phantomosaurus neubigi (Maisch & Matzke, 2006). This well-developed articular surface demonstrates that Besanosaurus also retained a functional basicranial articulation, although this process seems shorter in Besanosaurus than in Phantomosaurus. The caudal profile of the parabasisphenoid is characterized by a pair of caudolateral shallow natural notches and a slightly more evident median depression. On the ventral surface of the parabasisphenoid there is a clear bulge; this becomes more developed rostrally, where it tapers and becomes the cultriform process. In GPIT 1793/1, to the left side of this process, two carotid foramina are clearly visible; the cultriform process separates the carotid foramina as in Callawayia neoscapularis (ROM 41993, personal observation), Guizhouichthyosaurus tangae (IVPP V11853, personal observation), Macgowania janiceps (TMP 2009.121.1, personal observation) and Temnodontosaurus (Maisch, 2002), but differs from parvipelvians, in which a single carotid foramen is present (e.g., Maisch & Matzke, 2000).

Interestingly, PIMUZ T 1895 exposes a parasphenoid element, which forms a single ossification, not co-ossified with the basisphenoid. In fact, judging by the relative size of the specimen, PIMUZ T 1895 is osteologically less mature, and therefore likely ontogenetically younger than GPIT 1793/1. In PIMUZ T 1895 the cultriform process is preserved as a mediolaterally thin, median process pointing downwards; paired foramina for the carotid artery are present at its base, and the lateral extensions represent the basipterygoid processes.

We did not find any isolated basisphenoid in the examined specimen. A putative basisphenoid was mentioned by Maisch & Matzke (1997a) as present in GPIT 1793/1 as a separate ossified element, lying adjacent to the complete parasphenoid described above. It was originally described as having a “grossly triangular shape with a narrow anterior extension, and two lateral flanges (that) reminds of the general outline of the basal plate of the parasphenoid, being distinctly smaller, however”. We interpret this element as a broken piece of the basioccipital exposed in rostroventral view, possibly showing the basisphenoid facet.

Palatoquadrate complex

The palatal bones are preserved in between the flattened dermal skull elements in most of the specimens examined here. Some palatal elements are detectable through CT scans in BES SC 999, although the resolution is not good enough for us to describe their morphology. Therefore, the anatomy of the vomers, the palatines, and the pterygoids is based on specimens GPIT 1793/1, BES SC 1016 and PIMUZ T 4748 (Figs. 6, 8 and 10). These specimens are disarticulated and likely incomplete but do preserve palatal elements. For this reason, the reconstruction we propose (see “Cranial reconstruction” section below) is also based on Mixosaurus (Von Huene, 1916; Riess, 1986: fig. 5), Stenopterygius (McGowan & Motani, 2003: fig. 40), as well as personal observation of ‘Callawayia’ wolonggangense (SPCV 10305; Fig. S5) and Guizhouichthyosaurus tangae (IVPP V11853).

Vomer. In the holotype of B. leptorhynchus the vomers are not clearly detectable through CT scans. Possible vomers are exposed and disarticulated in PIMUZ T 4748 and GPIT 1793/1, in which they appear to constitute long, slender, and thin sheets of bone, slightly thicker caudally and pointed rostrally. In vivo, the vomers are partly articulated with each other, with the premaxillae and with the nasals, via long sutures that are partly preserved in GPIT 1793/1. In fact, in the right vomer exposed in lateral view the dorsal margin (partly covered) bear a thin vertical lamella of bone, which likely contacted similar median vertical processes of the other rostral bones. The lateral surface of the bone is concave all along its length. As noted by Maisch & Matzke (1997a), the rostral third of the ventral margin is reinforced by a ridge, whereas caudally another prominent ridge possibly represents the lateral facet for the palatine.

Palatine. In the holotype BES SC 999, a portion of the putative left palatine emerges within the left orbit, and a big portion of the right palatine is detectable through CT scans. However, these cannot be unambiguously identified. In GPIT 1793/1 we confirm the presence of two small splinters of bone, which lie adjacent to the elongate plate of the ?left vomer Maisch & Matzke (1997a), possibly representing incomplete portions of both palatines. In PIMUZ T 4748, possible fragmentary palatines are disarticulated from the other elements of the skull.

Pterygoid. The paired pterygoids are well-preserved in BES SC 1016, still semi-articulated and exposed in ventral view, separated caudally by the interpterygoid vacuity (Figs. 8 and 15). The left pterygoid is also partly preserved in 3D in GPIT 1793/1, where it is exposed in dorsal view, with the medial border of the rostral half partly folded onto the lateral portion of the bone and overlain by the right splenial. Remarkably, the pterygoid of GPIT 1793/1 also preserves the ascending process for the epipterygoid, which runs rostrocaudally, for at least the caudal half of the bone. Possible pterygoids are also preserved, albeit quite damaged, in specimen PIMUZ T 4748. As in other taxa, the pterygoid of Besanosaurus can be divided into two different halves. The rostral half tapers rostrally into a pointed tip, whereas caudally it becomes mediolaterally expanded, until reaching its maximum mediolateral width at its caudal end. The lateral margin of this second half hosts a wide concavity, in which the adductor mandibulae muscular complex would have been attached. In the caudalmost half of the pterygoid, three processes radiate in three different planes (see also Maisch & Matzke, 1997a): a caudolateral quadrate process, a caudomedial small bony flange, and a medial vertically ascending ridge that contacted the epipterygoid.

Figure 15 CT image of BES SC 1016.

CT image of specimen BES SC 1016. Note the pterygoids still in articulation and the relatively narrow interpterygoid vacuity (black arrow). The caudal extremities of the pterygoids are marked by grey arrows (full extent of the pterygoids is visible in Fig. 9). Note also the implantation of the mesialmost (caudalmost) premaxillary teeth, that are nested in separate sockets (white arrow). Scale bar represents 10 cm.

Unlike in Mixosaurus (SMNS 15378, personal observation; Maisch & Matzke, 1997b), the two pterygoids were not in close contact caudally in B. leptorhynchus, but slightly diverged from the midline leaving a narrow interpterygoid vacuity, which was definitely smaller than the wide opening observed in Guizhouichthyosaurus tangae (IVPP V 11853, Shang & Li, 2009). The median bulge of the parasphenoid may have emerged ventrally through this opening.

Epipterygoid. We agree with the interpretation of the epipterygoid given by Maisch & Matzke (1997a) for GPIT 1793/1. In this specimen, the epipterygoid is still articulated with the left pterygoid, although taphonomically compressed and slightly dislocated on its dorsal surface. The epipterygoid ventrally contacted the pterygoid with a large base that becomes much thinner (rostrocaudally reduced and mediolaterally very slim) approaching the dorsal end. The dorsal tip of this bone is missing, likely not preserved, although there is no apparent epipterygoid facet on the parietal roof. The presence of an ossified epipterygoid is significant as so far it was only reported in one other ichthyosaur: Ichthyosaurus (McGowan, 1973). The 2D nature of most preserved ichthyosaur specimens and the internal position of the epipterygoid hinders its identification in ichthyosaurs in general.

Quadratojugal. Putative quadratojugals are visible in the holotype of B. leptorhynchus BES SC 999. The right one is quite compressed and poorly preserved; the left one (formerly thought to be the left postorbital) lies on the rostral half of the left quadrate and is more complete. The only specimen in which the anatomy of the quadratojugal is clear is GPIT 1793/1 (Maisch & Matzke, 1997a; Fig. 16). In this specimen, the right quadratojugal is fully disarticulated and exposed in medial (internal) view. It has the shape of a trapezoidal plate with slightly convex cranial and caudal margins, and with a large flat triangular process that might have intruded dorsocranially between the postorbital and the squamosal, as in Temnodontosaurus (Maisch & Matzke, 1997a), likely contacting also the caudal end of the jugal (see also jugal). At the other end, i.e., in the caudoventral direction, the quadratojugal terminates in a prominent and thickened process, clearly offset from the main body of the bone and bearing a robust oblong articular facet for the quadrate. This facet, directed caudomedially, is similar to that in Phantomosaurus neubigi (Maisch & Matzke, 2006).

Figure 16 Jugal and quadratojugal of GPIT 1793/1.

Specimen GPIT 1793/1, left jugal in medial view (top) and right quadratojugal in lateral view (bottom). Dashed lines indicate portions of missing elements preserved as counterprints, thin lines indicate bone depressions. Abbreviations: see text. Scale bar represents 5 cm.

In articulated skulls, the quadratojugal was almost entirely overlapped by the squamosal, emerging laterally only with its stout quadrate process (Fig. 17) that recalls the morphology of Guizhouichthyosaurus tangae (IVPP V11869, personal observation) and Guanlingsaurus liangae (SPCV 03107, personal observation). The concave internal space left between the quadratojugal and the quadrate+pterygoid was likely occupied by the adductor mandibulae muscular complex (mAMES, mAMEM, mAMEP) (Jones et al., 2009).

Figure 17 Quadrates of Besanosaurus leptorhynchus.

Quadrates of Besanosaurus leptorhynchus. (A) right quadrate of PIMUZ T 4376 in rostrolateral (anterior) view; (B) CT scan of the left quadrate area of BES SC 999; (C) left quadrate of BES SC 999 in rostrolateral view, under visible light; (D) right quadrate of GPIT 1793/1 in rostrolateral view; (E) right quadrate of PIMUZ T 4847 in rostrolateral view; the upper half of the quadrate is missing, as well as most of its lateral portion. Interpretative drawings below. Dashed lines indicate portions of missing elements, thin lines indicate bone structures. Abbreviations: see text. Scale bar represents 5 cm.

As in many non-euichthyosaurian ichthyosaurs, the temporal region is proportionally long (Ji et al., 2016: char. 31.0). Despite this, compared to Guizhouichthyosaurus tangae (IVPP V11865, personal observation; Fig. S2), where most of the left squamosal is broken off, and the left quadratojugal is almost entirely exposed, the quadratojugal of Besanosaurus looks slender and proportionally smaller, although its general anatomy is not so different. In turn, the quadratojugal in Besanosaurus closely resembles the one of Shonisaurus (Nicholls & Manabe, 2004: fig. 6).

Quadrate. The left quadrate of BES SC 999 (partly hidden by the left postorbital) and the right quadrate of GPIT 1793/1 are preserved in rostrolateral view and are nearly identical; the incomplete ?right quadrate of PIMUZ T 4847 is also preserved but disarticulated (Fig. 17). Based on these specimens, the quadrate of B. leptorhynchus can be described as a relatively large element, kidney-shaped in outline, with a well-expressed lateroventral condyle for the articulation with the lower jaw and a stout dorsal head inserting in a proper “L-shaped” notch of the medial surface of the supratemporal vertical descending wall. The condyle is well developed on the rostrolateral side of the bone; just dorsal to the neck of the condylar process there is a rectangular articular facet for the quadratojugal. Along its concave lateral margin, the quadrate is thickened, whereas the convex side is tabular. On the medioventral border of the latter, there is a small triangular projection (“triangular process of the quadrate”, TPQ of Maisch & Matzke (1997a)) which has previously not been described in any other ichthyosaur, but we confirm it is also present in Guanlingsaurus liangae (SPCV 03107, personal observation; Fig. S3). This is the reason why we suggest to treat this character as a possible shastasaurid synapomorphy, rather than an autapomorphy of Mikadocephalus gracilirostris (see above). Medially, this process contacts the caudolateral edge of the pterygoid. In BES SC 999, the ‘triangular process’ of the right quadrate is located below the left ?quadratojugal (Figs. 10 and 17), and is therefore not visible on the surface but only through CT. This ‘triangular process’ may have had a structural-mechanical function, helping to hold in place, on its caudal side, the caudolateral flange of the pterygoid. On the caudomedial side, matching the anatomy of the adjacent skull elements, the quadrate likely bears a concave facet for the opisthotic, a small rounded and flat facet for the stapes, and a ventral large facet for the pterygoid, mediolaterally extending from the TPQ to the condyle for the lower jaw.

Mandible

Dentary. The dentary bones of Besanosaurus are well-preserved in BES SC 999 and PIMUZ T 4376, in which they are completely exposed in lateral view and still articulated. In both specimens the dentary is an extremely slender, narrow, elongate and approximately triangular bone that makes up almost three-quarters of the lower jaw length. The ventral margin of the dentary is straight rostrally and slightly concave caudally, whereas the dorsal (dentigerous) margin is straight all along its length. Each dentary contacts its contra-lateral element in a long symphysis mediorostrally, the surangular laterocaudally and the splenial mediocaudally, forming a pair of long oblique straight sutures that face ventrally. Both in BES SC 999 and PIMUZ T 4376, a narrow groove runs at mid-height on the lateral side of the dentary, from the caudal end towards the rostral tip, for four-fifths of the bone length. Above this mid-height line, the dorsal surface rostrally hosts a long dental groove and caudally a few teeth sockets (at least six). The labial shelf is partially exposed in PIMUZ T 4376, where it appears slightly concave. The 3D preservation of PIMUZ T 4376 also shows the aspect of the dentary-surangular suture, which is visible both on the lateral and the dorsal (buccal) surface of the lower jaw. This suture rostrally begins below the last five maxillary tooth positions and continues to the caudal end of the bone below the caudal margin of the lacrimal.

The very slender shape and the very gradual tapering towards the rostral tip of the dentary seen in Besanosaurus are similar to the dentary of ‘Callawayia’ wolonggangense (SPCV 10306; Chen, Cheng & Sander, 2007). On the other hand, contrary to Guizhouichthyosaurus tangae, the caudal end of the Besanosaurus dentary does not reach the midline of the orbit (Pan, Jiang & Sun, 2006), ranging from the rostral limit (PIMUZ T 4376, ?T 1895) to the rostral third of its length (BES SC 999). This slight variation may be due to ontogeny-related allometry, or to different preservation.

Splenial. This bone contributes to more than a half of the maximum height of the lower jaw, although in Besanosaurus the splenial looks less dorsoventrally expanded and straighter than in Ichthyosaurus (McGowan & Motani, 2003: fig. 41). It contributes to the medial (internal) wall of the lower jaw and is best observed in the holotype BES SC 999, where the lower jaw bones have been disarticulated and arranged in a fan-shape so that the caudal tip of the left splenial is almost entirely exposed. The splenial is long and slender and dorsoventrally taller than the angular. A small portion of the left splenial is visible, although largely obscured by adjacent elements, in PIMUZ T 4376.

The splenial contacts the dentary rostrally, the angular laterocaudally, the surangular laterodorsally, and a small portion of the prearticular caudodorsally. This is partially visible in medial view in the better exposed (right) splenial of GPIT 1793/1, and—to a lesser extent—in those of BES SC 999 and BES SC 1016. In all cases, the rostral end is thinner than the rest of the bone, and divided into two processes: a shorter, less robust dorsal one, and a longer, thicker ventral one. The latter possesses a slightly curved, concave facet that has been interpreted as the surface of contact for the dentary (Maisch & Matzke, 1997a), as in other taxa such as Ophthalmosaurus icenicus and lchthyosaurus (Andrews, 1910; Sollas, 1916). Both rostral tips of the two processes also show a rugose surface facing medially, which has been interpreted as contributing to the mandibular symphysis in Stenopterygius (SMNS 81961, personal observation) and Platypterygius australis (Kear, 2005). Given that this element is a left splenial exposed in medial view, we think this is also the case for GPIT 1793/1. The Meckelian groove of the splenial is represented by a long, slender, longitudinal depression, that gradually terminates rostrally in between the two processes.

Surangular. The lateral side of the articulated right and left surangular bones is well-visible in PIMUZ T 4376 and in the holotype BES SC 999. On the other hand, the disarticulated surangulars of GPIT 1793/1 and BES SC 1016 are exposed in medial view, and those of PIMUZ T 4847 are possibly exposed in medial view as well, although they are not much informative due to poor preservation. The surangular articulates with the dentary rostrodorsally and with the angular caudoventrally; caudomedially, the surangular is very often preserved in articulation with the articular.

In Besanosaurus the surangular is a robust and long bone, although shorter than the dentary. It possesses a relatively slender rostral process, whereas its main shaft becomes taller and thicker before reaching the coronoid (or preglenoid) process, its tallest point being at the level of the caudal end of the orbit. In medial view (GPIT 1793/1), a large elliptical neurovascular foramen opens here, at the center of the bone concavity, and a second much smaller foramen opens 1 cm caudal to it. In front and just caudal to the coronoid (preglenoid) process, the surangular of Besanosaurus narrows, expanding dorsoventrally again at its caudal end, in correspondence with the articulation with the articular bone.

Maisch & Matzke (1997a) described a separate coronoid in GPIT 1973/1, but we argue that a broken bone fragment was likely erroneously identified as the coronoid and that the coronoid in B. leptorhynchus was fused with the surangular, like in more derived ichthyosaurs. One specimen of the early-diverging ichthyopterygian Chaohusaurus brevifemoralis (Huang et al., 2019) has a separate coronoid on one side of the mandible, and a coronoid fused with the surangular on the other side. The coronoid in that specimen forms a distinct, pointed process in the caudal portion of the mandible - the coronoid process. In B. leptorhynchus (BES SC 999, PIMUZ T 4376) and all other more derived ichthyosaurs (see McGowan, 1973; Moon & Kirton, 2016) the mandible bears two distinct processes located in its caudal portion: a caudally located, pointed process - which corresponds in shape and position to the coronoid process in Chaohusaurus brevifemoralis (preglenoid process of Moon & Kirton, 2016), and a more dorsoventrally shallow process located rostrally to it—which Moon & Kirton (2016) described as the paracoronoid process. In light of the anatomy seen in Chaohusaurus brevifemoralis and the presence of two distinct mandibular processes in B. leptorhynchus, Ichthyosaurus and Ophthalmosaurus icenicus, we conclude that the coronoid likely became fused to the surangular in Merriamosauria (Ji et al., 2016), and formed a caudally located coronoid (preglenoid) process, with the surangular forming an additional, paracoronoid process, more rostrally.

The coronoid (preglenoid) process of Besanosaurus is thus remarkably large, distinctly rising above the dorsal margin of the surangular with a robust, rounded, and roughened dorsal tip, which probably served as attachment for an efficient mAMEM and mAMEP (Jones et al., 2009). The lateral wall of the surangular hosts a large longitudinal rough area, which is particularly well-visible in the holotype. This area is situated between upper and lower thicker and convex borders, resulting in a sligh depression on the lateral side of this bone; it likely served as the attachment of the externalmost mAMES, which was likely well-developed.

Interestingly, the caudal half of the surangular of B. leptorhynchus closely resembles that of Phantomosaurus neubigi (Sander, 1997), which has been recovered within Cymbospondylidae as the sister taxon to Cymbospondylus (Maisch & Matzke, 2000). Except for P. neubigi, no other ichthyosaur shows such a pronounced coronoid (preglenoid) process. However, since in many ichthyosaurs the jaws interlock tightly with each other, or are crushed on top of each other, this feature might be more widespread than currently established.

The anatomy of the surangular is more similar to ‘Callawayia’ wolonggangense (SPCV 10306; Chen, Cheng & Sander, 2007) than Guizhouichthyosaurus (Pan, Jiang & Sun, 2006), and also shows an intermediate condition between Cymbospondylus and more derived Merriamosauria.

Angular. The angular is preserved articulated with the rest of the lower jaw in PIMUZ T 4376 and disarticulated in GPIT 1793/1. In BES SC 999 the rostral tip of the angular occupies its original position, whereas the caudal end is displaced in a more ventral position, being taphonomically dislocated much like the splenial. In lateral view, the angular of Besanosaurus appears as a very elongate arched element that tapers rostrally to a gently curved, pointed end, with concave and convex dorsal and ventral margins respectively. Consequently, the angular forms its major contribution to the lateral side of the lower jaw in its caudal half, whereas rostrally the bone is overlapped by the surangular and forms the ventral side of the jaw. The contribution to the medial side of the lower jaw is represented by a low, long, rostrocaudally directed bony flange. The caudal end has a suboval outline in the sagittal plane, although it terminates with a straight suture for the articular. The angular is U-shaped in cross-section, contacting the surangular within its dorsal concavity, and the prearticular and the splenial medially. A well-preserved, three-dimensional facet for the surangular is exposed in medial view in GPIT 1793/1. The medial contact of the angular with the splenial is particularly thin and persists up to the rostral tip.

The overall shape if the angular, as well as the contacts with the adjacent bones, are more similar to ‘Callawayia’ wolonggangense (SPCV 10306; Chen, Cheng & Sander, 2007) and Guizhouichthyosaurus (Pan, Jiang & Sun, 2006), than to Ichthyosaurus, as pointed out for the latter by Maisch & Matzke (1997a).

Prearticular. In the best-preserved specimen (PIMUZ T 4376), the prearticular is situated at the caudal ends of the lower jaws, together with the articular. In the holotype, BES SC 999, the left prearticular is compressed and therefore larger than in life. In PIMUZ T 4847 a potential prearticular appears disarticulated from the surangular, but still partially articulated with the articular. In GPIT 1793/1, the prearticulars are fully disarticulated but fragmentary, both missing their rostral ends, and probably exposed in medial (internal) view. Maisch & Matzke (1997a) described them as thin, slender, crescent-shaped elements with a concave dorsal and convex ventral margin. Based on our observations of GPIT 1793/1 and all other specimens, we disagree with the statement on their “crescent-shape”, as the concavo-convex condition appears to be very faint, albeit present. When articulated, this bone contacted the angular caudoventrally and the articular medially, until its caudal end. In fact, the corresponding facets for the prearticular are preserved in the right prearticular of GPIT 1793/1. The prearticular of Besanosaurus looks very similar to that in other ichthyosaurs, such as Ichthyosaurus (McGowan, 1973) and Stenopterygius (personal observation), but it is different from Platypterygius australis, in which it is more angled caudodosally (Kear, 2005).

Articular. The articular is the most caudal and the most compact element of the lower jaw, forming the mandibular portion of the craniomandibular joint. This is reflected by its complex dorsomediolateral articular surface for the quadrate, rooted on a subrectangular base that is firmly sutured to the caudal end of the surangular (laterally) and of the prearticular (mediodorsally). This is best seen in PIMUZ T 4376 and GPIT 1793/1. The concavo-convex articular surfaces of the articular bone allow wide mobility around the condyle of the quadrate. The surface texture of the articular is also peculiar, being rather roughened and granular. As noted by Maisch & Matzke (1997a), the caudal positioning of the articular indicates that the retroarticular portion of the mandible was short.

As described for the holotype of Phantomosaurus neubigi by Maisch & Matzke (2006), a portion of the articular is exposed on the lateral surface of the lower jaw, posterior to the surangular, although this may be an effect of taphonomical compression in both cases.

Hyoid. Hyoid elements are visible in PIMUZ T 4847, and possibly BES SC 999. As in most vertebrates, the hyoid bones appear short if compared to the total length of the skull and lower jaw, but most of the hyoid apparatus is cartilaginous and thus not represented in fossils. If our identification is correct, the two symmetrical elements we see disarticulated in PIMUZ T 4847 are the first ceratobranchial pair, which are commonly preserved in the majority of ichthyosaurs as the only ossified hyoid elements (Motani et al., 2013). They are straight, rostrocaudally elongated, slender bones, a little shorter than the horizontal ventral ramus of the jugal. They are flattened mediolaterally and bear symmetrical subsquared rostral and caudal ends, that are slightly wider than the midshaft of the bone. This morphology can also be seen in the exposed caudal half of the possible right ceratobranchial I of BES SC 999. The first pair of ceratobranchials of Besanosaurus are very similar to those of Guanlingsaurus liangae (GNG dq-50; Ji et al., 2013) and Shonisaurus sikanniensis (TMP 94.378.2; Nicholls & Manabe, 2004) in being elongated, narrow in the middle, and broader at the extremities. The 3D structure is not preserved in the hyoids of Besanosaurus, so that they appear taphonomically straight.

Dentition

The dentition is visible in all examined specimens, although the most complete tooth rows are preserved in the holotype BES SC 999, in PIMUZ T 4376, in GPIT 1793/1, and in PIMUZ T 1895. Well-preserved premaxillary sockets are also visible in the CT of BES SC 1016 (Fig. 15). The teeth of B. leptorhynchus are typically small (if compared to the jaw length and width; File S4: char. 68), conical, without carinae, pointed, and widely spaced. The crown surface shows very fine apicobasal striations, which become more pronounced on the root surface, especially in the distalmost dentary teeth (PIMUZ T 4376; Fig. 18).

Figure 18 Teeth of Besanosaurus leptorhynchus.

Teeth of the specimens referable to Besanosaurus leptorhynchus. (A) PIMUZ T 4376: rostralmost (mesialmost) teeth of the dentary (above, turned upside down) and the premaxillae (below); (B) BES SC 999: premaxillary teeth at mid-length of the rostrum; (C) GPIT 1793/1: rostralmost teeth of the right maxilla; (D) PIMUZ T 4847: ?dentary teeth at mid-length of the rostrum. Scale bars represent 1 cm.

The premaxilla hosts at least 35 (estimated) teeth that interlock with the dentary teeth; the mesial-most premaxillary teeth are slightly curved, longer and more slender than the distalmost teeth, which are broader, shorter and with bulkier roots. The ventral surface of the premaxilla is flat and hosts several deep sockets, in which the teeth are implanted.

The maxillary teeth (11 preserved in PIMUZ T 4376) are shorter than the premaxillary teeth, and the distal-most ones are stouter with larger diameters but never blunt. The distalmost teeth are located almost below the caudal end of the external naris; the rostral (mesial) maxillary teeth are set in sockets, whereas the caudal (distal) teeth are set in a short groove, which is shorter than half of the rostral ramus of the maxilla as noted by Maisch & Matzke (1997a).

The dentary carries 38 (estimated) teeth, with the distalmost ones located below the rostral half of the rostral ramus of the maxilla. The dentary teeth show deeply striated roots, a feature that is more pronounced in the distalmost teeth. Also, the latter are much shorter and broader than the more mesial teeth (Fig. 18). In the dentary, the rostral (mesial) teeth are implanted in a long groove (PIMUZ T 1895), whereas the caudalmost (distalmost) teeth are set/implanted in deep sockets.

Discussion

Remarks on specimen size and intraspecific variation

The specimens we assigned to B. leptorhynchus can be ordered by size (skull length), from the smallest to the largest, as follows: PIMUZ T 4376 (405 mm), PIMUZ T 1895 (455 mm), BES SC 999 (522 mm), BES SC 1016 (530 mm), GPIT 1793/1(585 mm), PIMUZ T 4847 (710 mm). Table 2 reports the main cranial measurements, which were taken by all of the authors from the original fossils.

Table 2 Selected numbers and measurements.

Measure/number/ratio specimen	PIMUZ T 4376	PIMUZ T 1895	BES SC 999	BES SC 1016	GPIT 1973/1	PIMUZ T 4847	
1	Skull length: distance between tip of snout and caudal edge of articular surface of quadrate	405	[455]	522	[530]	[585]	(710)	
2	Jaw length: distance between tip of mandible and caudal edge of surangular	412	[475]	532	[550]	[600]	(743)	
3	Snout length: distance between tip of snout and rostral border of orbit	309	[320]	[366]	[368]	384	[590]	
4	Premaxillary length: distance between tip of snout and rostral tip of maxilla	184	[196]	214	[220]	/	/	
5	Prenarial length: distance between tip of snout and rostral border of external naris	224	[238]	270	[275]	/	/	
6	Snout ratio: snout length divided by jaw length	0,75	0,67	0,69	0,67	0,59	0,79	
7	Premaxillary ratio: premaxillary length divided by jaw length	0,45	0,41	0,40	0,40	/	/	
8	Prenarial ratio: prenarial length divided by jaw length	0,54	0,51	0,51	0,50	/	/	
9	Orbital diameter: maximum (rostrocaudal) internal diameter of orbit	67	[78]	[88]	[92]	[98]	[130]	
10	Orbital ratio: orbital diameter divided by jaw length	0,16	0,16	0,17	0,17	0,16	0,17	
11	Estimated number of sclerotic plates	[15]	[16]	[16]	/	/	[17]	
12	Internal diameter of the sclerotic ring (rostrocaudal)	21	[25]	[32]	/	/	[50]	
13	External diameter of the sclerotic ring (rostrocaudal)	51	[58]	[63]	/	/	[80]	
14	Internal diameter of the sclerotic ring (dorsoventral)	17	[26]	[28]	/	/	[45]	
15	External diameter of the sclerotic ring (dorsoventral)	38	[58]	[57]	/	/	[73]	
16	Parietal lenght along the midline	36	(45)	(48)	[50]	58	/	
17	Frontal length along the midline	49	55	(61)	/	71	[110]	
18	Lenght of caudal ramus of maxilla	(26,5)	38	[57]	(45)	(45)	/	
19	Length of rostral ramus of maxilla	81	83	[110]	(95)	138	/	
20	Lower jaw height at coronoid process	27	/	(59)	[57]	47	[69]	
21	Surangular height at notch for the jugal	25,5	/	(43)	42	41,5	[56]	
22	Length from the caudal margin of the naris to the rostral margin of the lower jaw	156	[220]	237	[300]	/	/	
23	Length of the cheek region	43	[60]	64	[75]	/	/	
24	Jaw depth at mid-length of the jaw	14	23	22	/	/	/	
25	Sclerotic ratio (sclerotic diameter 13 divided by orbital diameter)	0,31	0,32	0,36	/	/	0,38	
26	Mandibular ratio (jaw depth divided by jaw length)	0,03	0,05	0,04	/	/	/	
28	Overall body length	(2,124)	/	5,065	/	/	/	
29	Presacral length	[1,215]	[1,406]	2,052	/	/	(3,280)	
Note:

Selected numbers and measurements of each specimen of Besanosaurus leptorhynchus. Craniometric measures 1–10 are from McGowan & Motani (2003), 11–15 and 22–26 are from Dal Sasso & Pinna (1996), 16–21 are newly defined. Round brackets (*) indicate preserved but incomplete or deformed elements, square brackets [*] indicate estimated measurements of reconstructed elements.

Given that no significant qualitative and quantitative anatomical differences in the cranium are found between all specimens of B. leptorhynchus we examined, and given that the few discrepancies are probably due to taphonomical deformation, the measurements plotted in Table 2 likely directly correlate to size variation only. Therefore, we consider that the resulting signal indicates that all of the specimens of B. leptorhynchus examined here represent a possible ontogenetic series.

The skull length, together with the orbital diameter and the lower jaw length (Figs. 19B and 19C), increases with a constant slope through the possible ontogenetic stages. However, if compared to the presacral length (Fig. 19A), it appears that the body length grows much faster than the skull. This is more apparent once the animal has reached the reproductive ontogenetic stage, represented in Besanosaurus by the holotype BES SC 999.

Figure 19 Selected plots showing relevant cranial ratios across studied specimens.

Selected plots showing relevant cranial ratios across studied specimens supporting that they represent an ontogenetic series. (A) Presacral length/jaw length; (B) orbital rostrocaudal diameter/Jaw length; (C) jaw length/orbital rostrocaudal diameter.

Cranial reconstruction

Although the holotype of B. leptorhynchus (BES SC 999) preserves very little three-dimensional information, together with information from the referred specimens, it provides enough anatomical detail allowing us to infer most of the three-dimensional anatomy of the skull of the taxon. The reconstruction proposed in Fig. 20 depicts the bone proportions in the holotype, although much of the 3D anatomy has been inferred primarily from the best-preserved referred specimen (PIMUZ T 4376) and only a small portion of 3D anatomical information was obtained from other shastasaurid specimens (e.g., the arrangement of the palatal elements in ‘Callawayia’ wolonggangense (SPCV10305); Fig. S5). The unpreserved osteological features, represented in Fig. 20, were inferred following the methodology proposed by Bryant & Russell (1992), i.e., based on the cladistic distribution of known features in related taxa. A 3D 1:1 model of the skull was first reconstructed from thin cardboard based on the 2D drawings of the specimens (Figs. 4–9) and used as a reference for the main spatial distribution of the bones.

Figure 20 Cranial reconstruction of Besanosaurus leptorhynchus.

Cranial reconstruction of Besanosaurus leptorhynchus. Articulated skull and mandible in (A) left rostrolateral, (B) caudal (occipital), (C) dorsal, (D) left lateral, and (E) ventral (palatal) view. Abbreviations: see text. Line drawings by Marco Auditore.

Shapes, dimensions, and proportions of the skull roof bones are mostly based on PIMUZ T 4376. The morphology of the internal side of these bones is very well expressed in GPIT 1793/1.

The distinct curvature of the nasal, separating the gracile rostrum from the rest of the skull is inferred by the presence of extensive fractures in this region, observed in BES SC 999, PIMUZ T 4376 and PIMUZ T 1895, which, in particular, also shows similar additional fractures laterally, pointing out the presence of a short descending bony flange of the nasal, dorsally to the naris.

The proportions of the sclerotic ring are those of the holotype, however the anatomy of scleral plates is deducted from PIMUZ T 4376 and PIMUZ T 1895, since in BES SC 999 only half of the ring is preserved.

The lower jaw of BES SC 999 is unnaturally expanded dorsoventrally and the caudal elements have been partly disarticulated; to reconstruct the correct height and arrangement of the lower jaw, the model in Fig. 20 was based on the articulated lower jaw of PIMUZ T 4376.

The jugal turned out to be one of the most difficult bones to rearticulate with the rest of the skull. This bone is often found completely or partially disarticulated in the specimens described, and in any case, its caudal ramus is always loose from the adjacent bones. Based also on its remarkable length, we assumed that the caudal end of the jugal was simply juxtaposed, unsutured, medially to the squamosal and the postorbital, as seen also in Guizhouichthyosaurus tangae (IVPP 11896, personal observation) (Fig. S2).

The postorbital region is longer than tall in Besanosaurus and other shastasaurids, with the exception of ‘Callawayia wolonggangense’, in which the postorbital region is short, like in other post-Triassic ichthyosaurs.

The reconstruction in caudal view is based on BES SC 999, for the basioccipital; PIMUZ T 4376, which preserves the facets for the exoccipital on the basioccipital, and the quadrate; and GPIT 1793/1, which possesses the best preserved supraoccipital and opisthotic.

The 3D anatomy of the supratemporal and the way the quadrate articulates with the adjacent bone elements are inferred from the elements that articulate with them and based on comparison with a referred specimen of Guanlingsaurus liangae (SPCV 03107, personal observation) (Fig. S3).

The palatal view was based on BES SC 1016, which preserves both pterygoids; GPIT 1793/1, that shows a finely 3D preserved left pterygoid and putative palatines; and through observation of Guizhouichthyosaurus tangae (IVPP V 11853, personal observation) and ‘Callawayia’ wolonggangense (SPCV 10305, personal observation) (Fig. S5). The reconstruction we propose in Fig. 20E is also based on Von Huene (1916), Riess (1986: fig. 5), and McGowan & Motani (2003: fig. 40) for the general anatomy of the skull.

Phylogeny

In order to test the phylogenetic placement of B. leptorhynchus within Ichthyosauromorpha, we used the phylogenetic matrix of Huang et al. (2019), the most recently updated version of the phylogenetic matrix of Ji et al. (2016), a comprehensive dataset of 218 morphological characters scored for 73 ichthyosauromorph taxa. Even though the recently published character-taxon matrix of Moon (2017) is broader in scope, containing 287 characters scored for 116 taxa, we decided the matrix of Huang et al. (2019) was a more appropriate basis for performing a phylogenetic analysis as it focusses on non-parvipelvian ichthyosaurs. Our decision was motivated by the fact that Moon (2017) scored only 25 taxa (including only 3 non-parvipelvian taxa) (~22%) based on personal observation of fossil specimens, whereas Huang et al. (2019) scored the majority of taxa based on personal observation of relevant specimens, including all of the Chinese and American shastasaurids, and almost all Triassic ichthyosauromorphs in general (see Ji et al., 2016; Jiang et al., 2016; Motani et al., 2017).

B. leptorhynchus was scored on the basis of personal observation of the type and all referred specimens described in this study. In addition, the scorings of several cranial characters modified for the following shastasaurids, based on personal observation of fossil specimens, aided with relevant literature: Guizhouichthyosaurus tangae (IVPP V11865, IVPP V11869; Maisch et al., 2006b), ‘Callawayia’ wolonggangense (SPCV 10305, SPCV 10306; Chen, Cheng & Sander, 2007), Guanlingsaurus liangae (SPCV 03107; Sander et al., 2011) and Shastasaurus pacificus (UCMP 9017; Sander et al., 2011). All characters were treated as unordered and carrying equal weights.

The modified phylogenetic matrix (File S4) was analysed in TNT 1.5 (Goloboff, Farris & Nixon, 2008; Goloboff & Catalano, 2016), with memory set to hold 99,999 trees. The New Technology search option (a combination of Sectorial Search, Ratchet, Drift, and Tree fusing, with 100 random addition sequences) was used, followed by a round of TBR branch-swapping. Bremer support values were calculated in TNT 1.5 using the built-in Bremer Support tool from trees resulting from TBR branch swapping, by holding trees suboptimal by ten steps.

The analysis resulted in 14,480 most parsimonious trees (MPTs) of 713 steps (CI = 0.363, RI = 0.788). The phylogenetic analysis caused a loss of resolution at the base of Merriamosauria (last common ancestor of Shastasaurus pacificus and Ichthyosaurus communis and all of its descendants; Ji et al., 2016), with a strict consensus topology recovering a polytomy formed by all genera recovered in a monophyletic Shastasauridae in the analysis of Huang et al. (2019), Californosaurus perrini, Callawayia neoscapularis, Toretocnemidae and Parvipelvia. In 60% of the MPTs, B. leptorhynchus was recovered as the earliest-diverging representative of a ‘shastasaurid’ grade, but other possible resolutions of the Merriamosauria node recovered B. leptorhynchus as either the most basal taxon within a monophyletic Shastasauridae (as in Huang et al., 2019), or in a shastasaurid sub-clade comprising ((B. leptorhynchus, Guizhouichthyosaurs tangae), ‘Callawayia’ wolonngangense) (Fig. 21).

Figure 21 Cladogram of Ichthyosauriformes and phylogenetic position of Besanosaurus.

(A) 50% Majority rule consensus of 14,480 MPTs of 713 steps (CI = 0.363, RI = 0.788) obtained from parsimony analysis of the character-taxon matrix of Huang et al. (2019). Note that ‘shastasaurids’ are recovered as a grade at the base of Merriamosauria. Numbers above nodes indicate proportion of MPTs with specific node resolution. (B) Alternative resolution of Merriamosauria, with Shastasauridae recovered as a clade, and Besanosaurus leptorhynchus being the sister taxon of Guizhouichthyosaurus. (C) Alternative resolution of Merriamosauria, with Shastasauridae recovered as a clade, and Besanosaurus leptorhynchus being the earliest-diverging shastasaurid. Abbreviations: Call., Callawayia; Ch., Chaohusaurus; Cymb., Cymbospondylus; Lepto., Leptonectes; Mixo., Mixosaurus; Oph., Ophthalmosaurus; Qian., Qianichthyosaurus; Phal., Phalarodon; Shon., Shonisaurus.

The majority (60%) of MPTs resulting from our analysis recovered ‘Shastasauridae’ as a paraphyletic group at the base of Merriamosauria, which is in agreement with some other large-scale studies of ichthyosaur phylogeny that also recovered ‘shastasaurids’ as a grade (Callaway, 1989; Maisch & Matzke, 2000; Sander, 2000; Fröbisch et al., 2013; Moon, 2017). However, the remainder of the resulting MPTs recovered a monophyletic Shastasauridae, similar to the results obtained by Huang et al. (2019), Motani et al. (2017), Jiang et al. (2016), Ji et al. (2016) and Motani (1999). As a consequence, we do not consider the results of our analysis as conclusive for solving the controversy around shastasaurid monophyly/paraphyly. Because the introduction of new phylogenetic data from B. leptorhynchus and the revision of several cranial character states for other shastasaurids in the dataset of Huang et al. (2019) caused a loss of phylogenetic resolution at the base of Merriamosauria, it is expected that revisions of shastasaurid postcranial character scores, as well as the revision of character scores for other Triassic ichthyosaurs in general, might result in further changes to the topology obtained by Huang et al. (2019). Similarly, the relative phylogenetic position of B. leptorhynchus within Shastasauridae remains ambiguous, as in the subset of MPTs which recovered a monophyletic Shastasauridae, B. leptorhynchus was recovered as either the earliest-diverging shastasaurid, or forming a clade with the Ladinian–Carnian Chinese taxon Guizhouichthyosaurus (e.g., Maisch et al., 2006b; Jiang et al., 2020) and the Carnian Chinese taxon ‘Callawayia’ wolonggangense. Further studies of the taxonomy and phylogenetic relationships of Triassic ichthyosaurs are needed in order to unambiguously resolve the relationships at the base of Merriamosauria, but the results of our phylogenetic analysis, which recovered B. leptorhynchus as an early-diverging merriamosaurian, confirm its importance for our understanding of the early evolutionary and biogeographic history of shastasaurids, in particular, and merriamosaurians, more generally.

Feeding ecology of Besanosaurus

Ichthyosaurs appeared in the Early Triassic, just after the Permo-Triassic extinction event, and by the Middle Triassic they achieved a great taxonomic and ecomorphological diversity (Scheyer et al., 2014; Stubbs & Benton, 2016). The diversity in ichthyosaur taxa in the Besano Formation is a good example of this event, even in a relatively small marine basin. Indeed, we may assume the establishment of a niche partitioning between the ichthyosaurian taxa belonging to the Besano fauna: mixosaurids (with different ecologies), Cymbospondylus buchseri, and Besanosaurus leptorhynchus. Direct dietary evidence exists for Mixosaurus and Cymbospondylus. The holotype of Cymbospondylus buchseri shows a gastrointestinal content consisting exclusively of hooklets of soft-bodied coleoid cephalopods (Rieber, 1970; Sander, 1989).A recent case of dietary preference has been described for Mixosaurus cornalianus (Renesto et al., 2020), where the authors found tiny coleoid hooklets in the gut content of one specimen (BES SC 1000), which also preserves tiny fish vertebrae and scales from at least three different taxa. Hooklet dimensions suggest a partitioning driven by the size of the prey items in these two groups, and the additional presence of small fish in the Mixosaurus diet strengthens the idea that the two taxa relied on different food sources. In addition, mixosaurids and Cymbospondylus often seem to occur together, and with almost a global distribution (Nevada, Svalbard and Monte San Giorgio). This may explain how these two ichthyosaur taxa could share the same open marine environment. In fact, following the conclusions of Renesto et al. (2020), Mixosaurus cornalianus was likely an efficient open water swimmer and maneuvering exploiting BCF (body/caudal fin) periodic propulsion, and thus able to coexist with the bigger Besanosaurus and Cymbospondylus, occupying its own niche.

C. buchseri has been considered an apex predator (Fröbisch, Sander & Rieppel, 2006; Pardo-Pérez, Kear & Maxwell, 2020). However, its known gut contents show that this animal could fed lower in the food web, although this “last meal” does not exclude the possibility of a larger prey selection, consistently with its skull anatomy and tooth morphology.

The skull of an adult Besanosaurus leptorhynchus appears quite small, if compared to the overall body length, even smaller than in Guizhouichthyosaurus tangae (Pan, Jiang & Sun, 2006; Shang & Li, 2009). Jiang et al. (2020) recently described a thalattosaurian trunk in the stomach region of a Guizhouichthyosaurus, inferring macropredation in this taxon. The authors further hypothesized this behaviour for other large-bodied shastasaurids, including Besanosaurus. Given its longirostrine morphology, it is unlikely that Besanosaurus was a macropredator sensu Jiang et al. (2020). The rostrum of B. leptorhynchus is remarkably long and slender and equipped with several almost homodont, relatively tiny teeth, ornamented by faint grooves running from the tip to the base of the crown. The small skull and slender rostrum would have allowed rapid lateral or vertical movements of the head and would have been fairly hydrodynamic (at least in the cranialmost portion of the body), as suggested, e.g., for teleosaurids (Pierce, Angielczyk & Rayfield, 2009). Dal Sasso & Pinna (1996) reported that the teeth of Besanosaurus are more similar to that of Jurassic ichthyosaurs, such as Stenopterygius (see for example Dick & Maxwell, 2015: fig. 1), than to other Triassic ichthyosaurs known at that time, and hypothesized that Besanosaurus fed nearly exclusively on small coleoid cephalopods. The minute teeth and slender rostrum suggest that Besanosaurs was a “soft-prey specialist” sensu Fischer et al. (2016). Also, following Massare (1987) and Pierce, Angielczyk & Rayfield (2009), the mesial (rostral) teeth of Besanosaurus can be defined as the “pierce 1 guild” (long, delicate, and sharply pointed teeth for piercing small fish and soft cephalopods), and the distal (caudal) teeth as the “smash guild” (teeth bearing rounded points for grasping belemnoids and soft cephalopods), with the middle teeth showing an intermediate condition. This morphology and size do not change with skull and specimen size increase, unlike what was observed in Stenopterygius (Dick & Maxwell, 2015; Dick, Schweigert & Maxwell, 2016).

A single coleoid cephalopod hooklet surrounded by other gastric material is preserved in the thoracic region of the holotype of B. leptorhynchus, positioned 215 mm caudal to the right coracoid and 55 mm ventral to the vertebral column (Fig. 22). The hooklet is 2 mm long, has a robust shaft and a very curved uncinate tip, a central ridge, and a long articular process at the base, similar to the type C and D hooklets described by Pollard (1968: fig. 2). Among the known Middle Triassic (Monte San Giorgio) forms, the hooklet found in Besanosaurus is consistent in shape (but twice the size) with a mid-arm element of a specimen of Phragmoteuthis ?ticinensis (Rieber, 1970: figs. 1–3). Comparable hooklets are those recently found in the stomach region of a Mixosaurus cornalianus (Renesto et al., 2020: fig. 7A). In modern cetaceans, a long and slender rostrum with several small conical and homodont teeth is often coupled with a raptorial snap feeder-like hunting strategy, associated with a mostly piscivore diet (e.g., Stenella longirostris, Pontoporia, Inia and Platanista; Marx, Lambert & Uhen, 2016; Marshall & Pyenson, 2019). A longirostrine skull enables high velocity at the jaw tips, at the expense of biting forces, which is advantageous for capturing small elusive prey if combined to a rapid snapping of the jaws and fast lateral or vertical movements of the rostrum. Besanosaurus could fall into this category.

Figure 22 Hooklet of a coleoid cephalopod.

Isolated hooklet of a coleoid cephalopod preserved in the thoracic region of BES SC 999, holotype of Besanosaurus leptorhynchus. Scale bar represents 1 mm.

The inferred presence of well-developed jaw muscles (see surangular description) leads us to assume an efficient and fast jaw-closing movement, which is consistent with this hypothesis. That these muscles could have also generated strong biting forces is unlikely, given the thin and slender anatomy of the rostrum.

The convergence of a longirostrine skull morphology between diapsids and cetaceans (gharials and river dolphin, which primarily feed on small fishes) has been discussed by McCurry et al. (2017), who proposed that this convergence is driven by prey morphology and the methods of prey capture. Longirostrine morphology in river dolphins is also associated with a large degree of movement in the cervical vertebrae, more than in other oceanic species (Cassens et al., 2000; McCurry et al., 2017).

Piscivory for Besanosaurus would be further consistent with a prey driven niche partitioning among the Besano Formation ichthyosaurs. In any case, it appears that Besanosaurus would have preferred small, soft-bodied, and elusive prey which may have included coleoids. Unlike Besanosaurus, Cymbospondylus may have used a more forceful feeding strategy (slower feeding cycle and higher biting forces), given its less slender and more robust rostrum. Aside from the nature of the prey, it is clear that different hunting strategies, given by very different skull morphologies and dimensions, may have helped to maintain low competition among such a diverse ichthyosaur fauna, cohabitating the same ecosystem. Interestingly, in the southeastern Chinese Carnian faunas, in absence of a large cymbospondylid, the Shastasauridae diversity (three taxa with very different rostrine morphology and ecologies; see for example Chen, Cheng & Sander, 2007; Sander et al., 2011; Jiang et al., 2020) is greater than in the Besano Fauna.

If all three ichthyosaur taxa from the Besano Formation preyed on coleoids, we would expect a more abundant number of specimens and/or diversity in dibranchiate taxa. Such a diversity—also in term of ontogenetic stages—is not reported in the fossil record, but the lack of coleoids in the Besano Formation (with the exception of those in the gut contents) could be explained by particular bottom conditions of the basin not allowing the preservation of the cephalopod soft tissue.

To date, the Anisian record of a true ichthyosaurian apex predator in the form of a Thalattoarchon saurophagis-like animal (Fröbisch et al., 2013) is missing in the Besano Formation or in the broader Tethys realm. Cymbospondylus might have filled this gap, nevertheless no specimens larger than C. buchseri holotype (~5.5 m; Sander, 1989) have been reported from the Besano Formation to date. Nothosaurus giganteus might have had this role as well, however very few specimens have been reported from the middle part of the Besano Formation (e.g., PIMUZ T 4829), whereas most specimens are from the Muschelkalk deposits of the Germanic Basin (e.g., Rieppel, 2000; Klein et al., 2015). This sauropterygian was therefore presumably a dweller of nearshore environments. The ecological role of the three known ichthyosaurian taxa of the Besano Formation is still not fully understood, especially their swimming style, and still under examination, pending a detailed description of the postcranial osteology of Besanosaurus (under study elsewhere).

Longirostry and large size in Besanosaurus

Besanosaurus represents the earliest known large-sized marine diapsid (~8 m) that acquired an extreme longirostrine skull morphology: prior to Besanosaurus, marine reptiles possessing a longirostrine morphology never reached such size (Fig. 23). The idea that a longirostrine skull ensures high velocity at the jaw tips, which is advantageous for capturing small elusive prey is also consistent with Pierce, Angielczyk & Rayfield (2009) discussion about the morpho-functional significance of the longirostrine anatomy in teleosaurids (e. g., Plagiophthalmosuchus gracilirostris, Johnson, Young & Brusatte, 2020). An increased skeletal mass in teleosaurids, exemplified by a low pneumaticity and the development of thick osteoderms, would have resulted in an increased high body inertia (Hua & De Buffrenil, 1996; Pierce, Angielczyk & Rayfield, 2009). The mentioned authors suggested that a high body mass would also have permitted the body to remain stationary during rapid and precise movements of the head and neck. If we apply this to Besanosaurus we can infer that a longirostrine morphology may have reasonably coevolved with a high body mass, granting great benefits to the fishing ability of this animal. In addition, as discussed in the previous paragraph, foraging lower in the food web, together with an increased feeding efficiency, could have contributed to the appearance of remarkable large body size. In fact, following the model of Ferrón, Martínez-Pérez & Botella (2018) on gigantism in active marine predators, an increased feeding efficiency should have led to a more efficient way of consuming metabolic resources, triggering the possibility to acquire bigger body sizes without shifting to a higher metabolic level. A similar increase of body size, triggered by the acquisition of a more efficient feeding mode (although very different from longirostry), occurred in Mysticeti (Marx, Lambert & Uhen, 2016) and possibly happened at a higher metabolic level than in ichthyosaurs (Ferrón, Martínez-Pérez & Botella, 2018). A similar process may have resulted in the appearance of Late Triassic truly giant forms (>20 m), such as Shonisaurus sikanniensis (Nicholls & Manabe, 2004). Interestingly, some giant penguins (e.g., Icadyptes, Clarke et al., 2007) do also show a remarkable longirostrine beak. Indeed, the acquisition of longirostry, enabling to reach large sizes, should not have been the only ecological pathway to gigantism.

Figure 23 Early and Middle Triassic ichthyopterygian heads possessing longirostry.

Simplified outlines of four different Early and Middle Triassic ichthyopterygian heads possessing a long and slender rostrum. Specimens are at the same scale. (A) Utatsusaurus hataii (UHR 30691, Motani, Minoura & Ando, 1998); (B) Grippia longirostris (PMU R445, Motani, 2000); (C) Mixosaurus cornalianus (BES SC 1000, Renesto et al., 2020); (D) Besanosaurus leptorhynchus (PIMUZ T 4847, this paper); (E) Cymbospondylus buchseri (PIMUZ T 4351, Sander, 1989). Scale bar represents 10 cm.

Undoubtedly, gigantism may have granted many other advantages, such as a reduced vulnerability against predators, and possibly a more efficient body heat preservation, that in turn could have represented itself an intrinsic factor promoting gigantism, fueling positive feedback.

Conclusions

In general, the specimens here described preserve and represent a remarkably complete cranial anatomy, so that Besanosaurus leptorhynchus now is among the best-understood Middle Triassic Ichthyosaur taxa to date. Our revision of the skull morphology of this taxon clarified long-standing controversies regarding its cranial anatomy and the taxonomy of shastasaurids from Monte San Giorgio. Based on this rich fossil material, we have demonstrated that Mikadocephalus gracilirostris (GPIT 1793/1) is a junior synonym of Besanosaurus leptorhynchus, providing evidence to refute previous hypotheses (Maisch & Matzke, 1997a, 2000; Maisch, 2010) about the co-occurrence of two different shastasaurid taxa (Besanosaurus and Mikadocephalus) in the Besano Formation.

The six specimens here described represent a potential ontogenetic series covering a certain size range of mainly adult and potentially subadult specimens (Fig. 19), ordered by increasing size as follows: PIMUZ T 4376, PIMUZ T 1895, BES SC 999, BES SC 1016, GPIT 1793/1, PIMUZ T 4847. An allometric growth signal, yet to be fully tested, has also been detected. Other sources of intraspecific variation such as sexual dimorphism, cannot be ruled out, however, partly due to the limited dataset. Here we also report evidence that Besanosaurus was the largest Middle Triassic ichthyosaur taxon of the Western Tethys since we confidently estimate a fully adult size of about 8 m for specimen PIMUZ T 4847.

Besanosaurus possesses an extremely long, slender, and gracile snout, representing an ecological specialization never seen before the Anisian in a large sized (~8 m) diapsid. The diagnostic, prominent coronoid (preglenoid) process of the surangular and a large rugose area for the attachment of the mAMES allow to infer the presence of well-developed jaw closing muscles, which likely had an important functional role: we assume an efficient and fast jaw closing movement and hypothesize a snap-feeder-like hunting strategy, with a specific preference for small and elusive prey (such as coleoids and/or small fishes). Among the ichthyosaurian Besano-Monte San Giorgio Fauna (Cymbopondylus, mixosaurids, and Besanosaurus), different hunting strategies, demonstrated by different morphologies and dimensions of the rostra, should have led to the maintenance of low interspecific competition (i.e., niche partitioning). We also hypothesize that the specialization represented by a longirostrine morphology might have been driven by prey preference and the methods of prey capture. Mixosaurus and Cymbospondylus show almost a global distribution; on the contrary, Besanosaurus is known only from the Besano Formation (Italy and Switzerland). A wider distribution of this genus is expected (and supported by McGowan & Motani, 2003: 135–136): it seems unlikely to us that Besanosaurus would be represented only in the Alpine Tethys realm.

Last but not least, the importance of Besanosaurus is not only given by the completeness and remarkable preservation of its remains, and its ecological role, but also by the key phylogenetic position occupied by the taxon in the ichthyosaurian phylogeny: our analysis, performed with a matrix that includes around 90% of unambiguous scores for B. leptorhynchus and revised scores for other Triassic taxa, shows that this taxon represents the basalmost member of shastasaur-grade ichthyosaurs.

Supplemental Information

Supplemental Information 1 Perinarial region of Guizhouicthyosaurus tangae.

Perinarial region of a 3D-preserved referred specimen of Guizhouicthyosaurus tangae (IVPP V11869) in left lateral view. For bone interpretation, see Li & You (2002: fig. 2). Scale bar represents 5 cm.

Click here for additional data file.

Supplemental Information 2 Postorbital region of Guizhouicthyosaurus tangae..

Postorbital region of a 3D-preserved referred specimen of Guizhouicthyosaurus tangae (IVPP V11865) in left lateral view. Abbreviations: see text. Scale bar represents 5 cm.

Click here for additional data file.

Supplemental Information 3 Postorbital region of Guanlingsaurus liangae..

Postorbital region of a 3D-preserved referred specimen of Guanlingsaurus liangae (SPCV 03107) in left lateral view. For bone interpretation, see Sander et al. (2011: fig. 2b).

Click here for additional data file.

Supplemental Information 4 Occipital region of Guanlingsaurus liangae..

Occipital region of a 3D-preserved referred specimen of Guanlingsaurus liangae (SPCV 03107) in caudomedial view. Note the presence of the triangular process of the quadrate, which helps to hold the caudolateral flange of the pterygoid. Abbreviations: see text. Scale bar represents 5 cm.

Click here for additional data file.

Supplemental Information 5 Palatal view of ‘Callawayia’ wolonggangense.

Palatal view of a referred specimen of ‘Callawayia’ wolonggangense (SPCV 10305). For bone interpretation, see Chen et al. (2007: fig. 2D).

Click here for additional data file.

Supplemental Information 6 Strict consensus tree of Ichthyosauriformes.

Strict consensus of 14,480 MPTs of 713 steps (CI=0.363, RI=0.788) obtained from parsimony analysis of the character-taxon matrix of Huang et al. (2019). Note large polytomy at the base of Merriamosauria. Numbers above nodes indicate Bremer support values. Abbreviations: Call., Callawayia; Ch., Chaohusaurus; Cymb., Cymbospondylus; Lepto., Leptonectes; Mixo., Mixosaurus; Oph., Ophthalmosaurus; Qian., Qianichthyosaurus; Phal., Phalarodon; Shon., Shonisaurus.

Click here for additional data file.

Supplemental Information 7 3D models of BES SC 999, PIMUZ T 4376, and GPIT 1793/1.

3D model obtained with photogrammetry of the skull and mandible of BES SC 999; 3D model obtained with photogrammetry of the skull and mandible of PIMUZ T 4376; 3D model obtained with photogrammetry of specimen GPIT 1793/1.

Click here for additional data file.

Supplemental Information 8 Data matrix used for the cladistic analysis.

Click here for additional data file.

We thank the volunteers of the former “Gruppo paleontologico di Besano”, who unearthed the holotype of Besanosaurus leptorhynchus, and many other exceptional fossils. We also thank F. Fogliazza (MSNM) and C. Egli (PIMUZ) for preparation of the two new skulls; M. Auditore for anatomical drawings; L. Forzenigo, C. Bonelli, and G. Terribile (Fondazione IRCCS “Cà Granda” Ospedale Maggiore Policlinico di Milano) for CT analysis of the whole holotype of B. leptorhynchus; the Soprintendenza Archeologica della Lombardia for permissions. For access to key specimens in museum collections we thank T. Goller and I. Werneburg (GPIT), R. B. Hauff (Urweltmuseum Hauff, Holzmaden), C. Klug (PIMUZ), E. Maxwell (SMNS), L. Cheng and X.-H. Chen (WGSC), C. Li and Q. Shang (IVPP), Pat Holroyd (UCMP), Kevin Seymour and Brian Iwama (ROM). We also thank M. Auditore, M. Balini, S. Maganuco and G. Teruzzi, for helpful discussions. This paper is part of a Ph.D. project (G. Bindellini) focusing on the Besano Formation. Fauna, led by the Università Statale di Milano (M. Balini) in agreement with the Museo di Storia Naturale di Milano (C. Dal Sasso). Finally, we thank the editor Mark T. Young, Neil Kelley and an anonymous reviewer for their constructive comments.

Additional Information and Declarations

Competing Interests

Author Contributions

Data Availability

Cristiano Dal Sasso is an employee of the Museo di Storia Naturale di Milano, Italy. Torsten Michael Scheyer is an employee of the Paläontologisches Institut und Museum, Universität Zürich, Switzerland.

Gabriele Bindellini conceived and designed the experiments, performed the experiments, analyzed the data, prepared figures and/or tables, authored or reviewed drafts of the paper, and approved the final draft.

Andrzej S. Wolniewicz conceived and designed the experiments, performed the experiments, analyzed the data, prepared figures and/or tables, authored or reviewed drafts of the paper, and approved the final draft.

Feiko Miedema conceived and designed the experiments, performed the experiments, analyzed the data, prepared figures and/or tables, authored or reviewed drafts of the paper, and approved the final draft.

Torsten M. Scheyer conceived and designed the experiments, performed the experiments, analyzed the data, prepared figures and/or tables, authored or reviewed drafts of the paper, and approved the final draft.

Cristiano Dal Sasso conceived and designed the experiments, performed the experiments, analyzed the data, prepared figures and/or tables, authored or reviewed drafts of the paper, and approved the final draft.

The following information was supplied regarding data availability:

The data matrix used for the cladistic analysis is available in the Supplemental Files.

3D photogrammetric scans are available at Morphosource:

- GPIT 1793/1: DOI 10.17602/M2/M350107

- PIMUZ T 4376: DOI 10.17602/M2/M346884

- BES SC 999: DOI 10.17602/M2/M350112

BES SC 999 is stored and accessible in the Museo di Storia Naturale di Milano, Milan, Italy.

PIMUZ T 1895, 4376 and 4847 are stored and accessible in the Paläontologisches Institut und Museum der Universität Zürich, Zurich, Switzerland.

GPIT 1793/1 is stored and accessible in the Palaeontological Collection of Tübingen University, Tübingen, Germany.

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
