# Peer review of "Cranial anatomy of Besanosaurus leptorhynchus Dal Sasso & Pinna, 1996 (Reptilia: Ichthyosauria) from the Middle Triassic Besano Formation of Monte San Giorgio, Italy/Switzerland: taxonomic and palaeobiological implications"

_PeerJ, doi:10.7717/peerj.11179_

## Round 0.1 · original submission · Minor Revisions

Dear authors,

I have accepted the reviewers decision of 'minor revisions'. Reviewer one made an interesting comment regarding the phylogenic matrix used, that you will need to address prior to re-submission. Reviewer two was unable to access the GoogleDrive with the 3D models, and they also made comments on the figures.

Longirostry and 'gigantism'. I'd be a little more cautious there. Yes, among extant crocodylians Tomistoma and Gavialis reach long body lengths, in terms of mass, Crocodylus porosus is the 'most massive'. It's worth noting that both Tomistoma and Gavialis are the largest predators in their riverine ecosystems. Be careful you aren't comparing "apples with oranges".

I look forward to receiving your revised manuscript.

Reviewer 1 ·

Basic reporting

This is a very well conducted, informative and important contribution to the field. .
The manuscript is principially well written but contains few spelling and gramma mistakes, missing or additional blanks.
Some paragraphs are flush left , some are justified.
sometime there is a blank after PIMUZ T 43.. sometimes not. PIMUZ T43...

Experimental design

well conducted and appropriate illustrated

Authors justify why they use the Motani matrix and stated why not using the Moon matrix
I am fine with this.
However, they should be aware that if the Motani et al. 2017 matrix is based on Ji et al. 2016 (which is likley) that this matrix contains some typos and other errors, when not corrected...
but maybe neglegible for this study which mainly wants to analyse the relationships of Besano.

Validity of the findings

This is a detailed, well conducted and important revision of skull anatomy of Besanosaurus.
The conclusion that Mikadocephalus is a junior synonym of Besanosaurus is well supported and was long overdue.

Additional comments

here are some specific comments
line
42: body size
75 (large-sized)
106 their ? which
175, indicating deep water ?
179, maybe replace features by preservation
you mention that the specimens can be ordered by size in line 192, so I am wondering which order you follow in your description from line 196 to 240
226, delete more
230, please check
what is the rec./estimated body size of GPIT 1793/1 and BES SC 1016
303, maybe replace bad by poor
319 ?and are almost complete
498-499 please rephrase
502, the contact of elements (instead of bone)
675 delete one .
1288- 1290 repetition to line 192
1291 unclear: every author measured every specimen ???
1333 please add taxon name for IVVP specimen
1409 this must be Fröbisch et al. 2006
1458 ?in other
1523-1524 repetition but maybe ok here/add blank
Fig. 10 please add to the Fig. legend whay you choose to show these slices/what is important here
Fig 11 please mark the naris with an arrow(s)
Fig. 12 please add to Fig Legend what exactly you want to show/point out here
maybe undlay the parts of the mx in the outline sketches with a light grey to further illustrate the shape
Fig. 13 ditto please shortly explain what you want to point out/visualize with the UV light
Fig. 15 please label or line pterygoids
Fig. 17 maybe mirrow B
Fig. 19 Fig Legend: Selected plots showing several cranial ratios across studied specimens supporting that the specimens represent an ontogenetic series
Fig. 20 I would move this nice reconstruction after the figs of single skulls (Fig. 10?) correct ichthyosauromorpha
Fig. 21 maybe move also cladogram forward

·

Basic reporting

Generally a very strong paper and an important contribution.

There are some minor issues in communication detailed below.

The 3D photogrammetry models were not accessible to me in the Google Drive without prior permission, but this is not a deal breaker.

Experimental design

No critical comments on this front. I appreciate the new descriptive data, figures and analysis of important specimens that have been in collections for many years but not been featured in prior technical publications.

Validity of the findings

The overall description of specimens is detailed seems solid. The phylogenetic analysis builds on previous analyses. Some of the macroecological/macroevolutionary speculation at the end is debatable, but is plausible and well-founded overall. Some suggestions are offered below to clarify this section in a few places.

Additional comments

Abstract.

Line 42 - ‘Clade’ is redundant with ‘monophyletic,’ can be omitted.

Line 42 - ‘Size’ is a bit ambiguous, consider ‘body length’ instead if appropriate.

Line 44 - Minor quibble but ‘never before seen’ seems like an odd characterization for a taxon known since 1996 … ‘never before reported’


Introduction

Line 49 - Consider replacing ‘first’ or ‘earliest’ for ‘oldest’ which can have ambiguous meaning.

The authors avoid discussing the large ichthyosaur remains of Svalbard (e.g. ‘Pessopteryx’) which extend downward to the Olenekian. At times some of these have been suggested to have affinities with shastasaurs, including Besanosaurus (e.g. McGowan and Motani, 2003), although this has been questioned and there is a complicated history and taxonomy. (See Maxwell and Kear 2015). I can understand avoiding this as there are contradictory interpretations in the literature, but it might warrant a brief mention?

Line 83 - I think ‘had’ should be ‘having’ to make grammatical sense.


Materials & Methods

Line 209 - Slight ambiguity here: “last preserved caudal vertebra (the distalmost caudals are missing or spaced apart).” Do the authors mean that they measured to the last caudal vertebrae preserved in articulation and ignored the ‘missing or spaced apart’ distal most caudals?

LIne 226 - omit the first, extraneous ‘more’

Line 230 - “Is Preserved’ should probably be “It is preserved” (note missing word & capitalization.

Line 230 - Suggest replacing “position, it” with “position. It” (new sentence).

Results

Line 490 - replace ‘results’ with ‘is’

Line 533 - Again, non-standard use of ‘results’ here to me and the sentence overall is a bit vague. Can you clarify re resemblance to Guizhouichthyosaurus, is the similarity in size or in shape (or both)? NB: Now I notice the colon so I guess the similarity is about the contribution of the frontal? Regardless this sentence needs to be rewritten for clarity.

Line 548 - Again, ‘results’ should probably be ‘is’

Line 610 - replace ‘result’ with ‘be’

Line 742 - replace ‘results’ with ‘is’

Line 791 - ‘e. g.’ Is inconsistently formatted throughout the paper, sometimes with a space (as here), with or without a comma. I think it should be “e.g.,” (no space and with a comma).

Line 1260 - ‘appear’ instead of result?

Line 1281 - Again, I think ‘ result’ should be ‘are’ — Also a bit unclear if you are comparing just the roots as ‘much shorter and broader’ or the overall tooth morphology here

Line 1299 - omit ‘shows to’

Lines 1298/1302 - Figure 19: Not placing the origin of these plots at zero will provide a distorted perception of the slope. Also, allometric relationships are MUCH easier assessed when the data is plotted on a log-log axis, ideally with a reference line for isometry (exponent =1)

Lines 1305 - replace “few” with “little”

Line 1329 - replace ‘resulted’ with ‘turned out’ or just ‘The juggle was one of the most difficult…”

Lines 1327 - 1376 - This information was all already given in methods above so can probably be omitted here.

Line 1419 - I’m not quite sure what ‘relevant’ means here?

Line 1445 - ‘associated with’ instead of ‘associated to’

Line 1446 - Note ‘Peyenson’ is misspelled here, should be ‘Pyenson’ Also I’d probably replace the colon with a period and start a new sentence with, “A longirostrine skull,” also I would say something like ‘facilitates’ or ‘enables’ rather than ‘ensures’ high velocity.

Line 1462 - I would reconsider the use of the term ‘Rammer-like’ here. “Ram-feeding” was coined initially by Liem 1980 to describe a feeding mode in teleost fish where prey are engulfed via coordinated acceleration of the body and mouth. It has later been applied to describe a variety of feeding modes in other aquatic vertebrates (not always consistently).

I think the authors are implying that Cymbospondylus would have used slower feeding cycle by higher bight forces to subdue larger but less evasive prey. This could involve a degree of ‘ram feeding’ but honestly the raptorial ‘snap-feeding’ suggested for Besanosaurus would also involve a form of ‘ram feeding’ sensu Liem 1980.

Perhaps rephrase as ’Cymbospondylus may have used a more forceful feeding strategy’ or something along those lines? I think that expresses the same idea without invoking a term that has some baggage.

Line 1474 - I don’t follow the reasoning entirely here, as anoxic conditions would be expected to generally favor soft-tissue preservation? In general though, soft-bodied coleoids do not have a fantastic fossil record so maybe the relative paucity isn’t so surprising?

Lines 1475 - 1481 - Nothosaurus giganteus might have filled this role in this ecosystem? Large macropredatory sauropterygians are scarce in the Anisian of North America apart from some fragmentary remains from British Columbia described by one of the co-authors.


Lines 1484-1509 - At first I was a bit skeptical of this scenario but I think the authors do a good job of articulating this, although it needs further investigation. Interestingly both Tomistoma and Gavialis attain very large sizes, among the largest extant crocodilians, which could support this scenario. Also, giant penguins (e.g. Icadyptes) are also extremely longirostrine so the authors might be on to something here.

Nevertheless, It seems to me that this is not the only ecologic pathway to gigantism in marine vertebrates. Deep diving & macropredatory ecologies are also associated with increased body sizes in many marine clades (pinnipeds, sharks & cetaceans).

Likewise there are numerous examples of longirostrine taxa that remain relatively small, among fish and aquatic/marine reptiles & amphibians. For example Mixosaurus cornelianus, which lived alongside Besanosaurus was rather longirostrine but much smaller. So the correlation between longirostry, low trophic level feeding & gigantism is a loose, and Besanosaurus likely represents one of many pathways toward gigantism/longirostry rather than the dominant mode.

Figure 3:

I find the uneven lighting/contrast across these panels to make it a bit difficult to get as much out of this figure as I would like. I know from experience how difficult it is to light and photograph large specimens particularly when they are on a slab and the contrast between bone and matrix is limited. I wonder if there might not be room for improvement, with some post-processing of brightness and lighting?
Alternatively some interpretive line drawings could be useful but I realize that is a lot of extra work.

Figure 19

See comments above.

Figure 21

It might be just the review PDF but the image quality here is a little low, although it does not hamper reading of the figure. The shading should probably be explained in the figure caption, or perhaps even better, labelling of key nodes.

Figure 23

I’m not sure this figure is adding much, it could probably be moved to a supplement. It might be most useful to see Besanosaurus & Cymbospondylus side by side as that comparison is emphasized more in the manuscript text. Why the gray shading on Grippia?

---

## Round 0.2 · Minor Revisions

Dear authors,

I've made the decision of 'minor revisions' after reading your revised manuscript. I noticed typos, in some scientific names, and some grammatical issues (particularly in some of the new text, and in some discussion sections and the conclusions). Please note that PeerJ does not provide a full linguistic check during proof-stage, so a final check through the text would be beneficial.

In the discussion, the teleosauroid information will need a minor update after the Johnson et al. (2020, PeerJ paper on teleosauroid taxonomy).

In the emend diagnosis, can you indicate which characters are autapomorphic?

These changes should be quick and easy changes to make, and I anticipate that your manuscript will be accepted.

---

## Round 0.3 · accepted · Accept

Dear authors,

Thank you for your revised manuscript, I have made the decision to 'accept'.

The production staff will shortly be in contact to take you through the proof stages.

Once again, thank you for choosing PeerJ, and I hope you will use us again in the future.